



# The Community Multiscale Air Quality (CMAQ) Model Versions 5.3 and 5.3.1: System Updates and Evaluation

K. Wyat Appel, Jesse O. Bash, Kathleen M. Fahey, Kristen M. Foley, Robert C. Gilliam, Christian Hogrefe, William T. Hutzell, Daiwen Kang, Rohit Mathur, Benjamin N. Murphy, Sergey L. Napelenok, Christopher G. Nolte, Jonathan E. Pleim, George A. Pouliot, Havala O. T. Pye, Limei Ran[a], Shawn J. Roselle[*], Golam Sarwar, Donna B. Schwede, Fahim Sidi, Tanya L. Spero, David C. Wong

Center for Environmental Measurement and Modeling, Office of Research and Development, U.S. Environmental Protection Agency, RTP, NC, USA

[a] now with: Natural Resources Conservation Service, U.S. Department of Agriculture, Greensboro, NC, USA
[*] Retired

*Correspondence to:* K. Wyat Appel (appel.wyat@epa.gov)

**Abstract.** The Community Multiscale Air Quality (CMAQ) model version 5.3 (CMAQ53), released to the public in August 2019 and followed by version 5.3.1 (CMAQ531) in December 2019, contains numerous science updates, enhanced functionality, and improved computation efficiency relative to the previous version of the model, 5.2.1 (CMAQ521). Major science advances in the new model include a new aerosol module (AERO7) with significant updates to secondary organic aerosol (SOA) chemistry; updated chlorine chemistry; updated detailed bromine/iodine chemistry; updated simple halogen chemistry; addition of dimethyl sulfide (DMS) chemistry in the CB6r3 chemical mechanism; updated M3Dry bi-directional deposition model; and the new Surface Tiled Aerosol and Gaseous Exchange (STAGE) bi-directional deposition model. In addition, support for the Weather Research and Forecasting (WRF) model's hybrid vertical coordinate (HVC) system was added to CMAQ53 and the Meteorology-Chemistry Interface Processor (MCIP) version 5.0 (MCIP50). Enhanced functionality in CMAQ53 includes the new Detailed Emissions Scaling, Isolation and Diagnostic (DESID) system for scaling incoming emissions to CMAQ and reading multiple gridded input emission files.

Evaluation of CMAQ53 was performed by comparing monthly and seasonal mean daily 8-hr average (MDA8) $O_3$ and $PM_{2.5}$ values from several CMAQ531 simulations to a similarly configured CMAQ521 simulation encompassing 2016. For MDA8 $O_3$, CMAQ531 has higher $O_3$ in the winter versus CMAQ521, due primarily to reduced dry deposition to snow, which strongly reduces wintertime $O_3$ bias. MDA8 $O_3$ is lower with CMAQ531 throughout the rest of the year, particularly in spring, due in part to reduced $O_3$ from the lateral boundary conditions (BCs), which generally increases MDA8 $O_3$ bias in spring and fall. For daily 24-hr average $PM_{2.5}$, CMAQ531 has lower concentrations on average in spring and fall, higher concentrations in summer, and similar concentrations in winter as CMAQ521, which slightly increases bias in spring and fall and reduces bias in summer. Comparisons isolating updates to several specific aspects of the modeling system, namely the lateral BCs, meteorology model version, and the deposition model used, were also performed. Transitioning from a hemispheric CMAQ (HCMAQ) version 5.2.1 simulation to a HCMAQ version 5.3 simulation to provide lateral BCs contributes to higher $O_3$ mixing ratios in the regional CMAQ simulation in higher latitudes during winter (due to the decreased $O_3$ dry deposition to snow in



CMAQ53) and lower O₃ mixing ratios in mid and lower latitudes year-round (due to reduced O₃ over the ocean with CMAQ53). Transitioning from WRF version 3.8 to WRF version 4.1.1 with the HVC system resulted in consistently higher (1.0 – 1.5 ppbv) MDA8 O₃ mixing ratios and higher PM₂.₅ concentrations (0.1 – 0.25 µg m⁻³) throughout the

year. Finally, comparisons of the M3Dry and STAGE deposition models showed that MDA8 O₃ is generally higher with M3Dry outside of summer, while PM₂.₅ is consistently higher with STAGE due to differences in the assumptions of particle deposition velocities to some surfaces between the two models. For ambient NH₃, STAGE has slightly higher concentrations and smaller bias in the winter, spring, and fall, while M3Dry has higher concentrations and smaller bias, but larger error and lower correlation, in the summer.

## 1 Introduction

To help protect human health and the environment, many countries and government organizations around the world have set limits for atmospheric pollutants. Established in 1970 and working under the direction of the Clean Air Act (CAA) of 1970 and revised CAA of 1990 (https://www.epa.gov/clean-air-act-overview), the United States Environmental Protection Agency (USEPA) is mandated to periodically review and propose revised limits, or national

ambient air quality standards (NAAQS; Bachmann, 2007), for criteria air pollutants in the U.S. including ground level ozone (O₃) and PM₂.₅ (particulate matter with an effective aerodynamic diameter less than 2.5 µm). The current U.S. NAAQS for ground level O₃ is 0.070 parts per million by volume (ppmv) and is based on the maximum of a rolling 8-hr average throughout the day (MDA8 O₃). The current U.S. NAAQS for PM₂.₅ is 35 µg m⁻³ for a daily average and 12 µg m⁻³ for an annual average. Areas above the NAAQS are considered to be in "nonattainment" and are required

to implement measures that reduce observed O₃ and/or PM₂.₅ to be below the NAAQS within a specified period of time. These measures typically require the reduction of anthropogenic emissions within the area of nonattainment, as both O₃ and PM₂.₅ are principally formed in the atmosphere through a complex series of chemical reactions from primary emitted pollutants. For O₃, these precursor pollutants are oxides of nitrogen (NO and NO₂) and volatile organic compounds (VOCs) such as isoprene and formaldehyde (HCHO), which undergo photochemical reactions in the

atmosphere to form ground level O₃. Ambient PM₂.₅ in contrast arises from both primary emissions and complex chemical reactions leading to formation of its diverse inorganic and organic chemical constituents. Therefore, a much larger system of emission sources and atmospheric reactions must be considered for PM₂.₅ formation. In addition to the formation of these pollutants, there are numerous processes that also destroy (e.g., titration, photolysis, oxidation) and remove (e.g., dry deposition, wet scavenging) these pollutants from the atmosphere.

Given the complexity of O₃ and PM₂.₅ formation, transport, destruction, and deposition, the groups (e.g., state governments, multi-jurisdictional organizations) responsible for developing plans to reduce these pollutants generally rely on numerical models to simulate the processes involved and estimate the outcomes of plausible control strategies. The Community Multiscale Air Quality (CMAQ) model (Byun and Schere, 2006), developed and distributed by the USEPA's Office of Research and Development, is a state-of-the-science numerical air quality model with

comprehensive representations of the emission, transport, formation, destruction, and deposition of many air pollutants, including O₃ and PM₂.₅. The first version of CMAQ was released more than 20 years ago and has been



continually updated to incorporate the latest available scientific information and improve overall model performance. CMAQ version 5.3 (CMAQ53; USEPA, 2019a) was released in August 2019 and was followed by a minor update in December 2019 to version 5.3.1 (CMAQ531, USEPA, 2019b). Prior to these two versions, the previous version of CMAQ was version 5.2.1 (CMAQ521; USEPA, 2018) released in July 2018. The most recently published comprehensive review and evaluation of CMAQ was conducted for version 5.1 (Appel et al., 2017), although numerous articles on specific scientific updates to CMAQ have been published since then.

Here we describe the scientific updates in CMAQ53, review several of the updates that improve simulation speed and provide enhanced capabilities, and present a comparison against the previous release (i.e. CMAQ521). Some of the major scientific enhancements in CMAQ53 include updates to aerosol formation, including monoterpene secondary organic aerosol (SOA) formation and uptake of water onto hydrophilic organic aerosol (OA); updates to the M3Dry deposition and bi-directional ammonia ($NH_3$) exchange model; the new Surface Tiled Aerosol and Gaseous Exchange (STAGE) deposition model; and updates to marine halogen and chlorine chemistry. Examples of enhanced capabilities in CMAQ53 include the new Detailed Emissions Scaling, Isolation, and Diagnostic (DESID) module providing the ability to scale emissions directly through a single control file; the updated Integrated Source Apportionment Method (ISAM; Kwok et al., 2013, 2015) for source tracking and attribution; and the updated Sulfur Tracking Method (STM). The operational evaluation of the new modeling system uses 2016 annual simulations covering the conterminous United States, along with large portions of Canada and northern Mexico. This evaluation quantifies the impacts of the scientific updates in CMAQ53 on model performance as compared to CMAQ521, while also analyzing the impacts that other specific system updates (i.e. meteorology, boundary conditions, and deposition model) have on model performance.

## 2 Review of scientific improvements in CMAQ version 5.3

Scientific updates in CMAQ are contributed by many different researchers, who in some cases may work for years to develop and test that update. This section primarily focuses on only the most impactful scientific updates in CMAQ53, determined by examining the impacts each change has on concentrations of $O_3$ and $PM_{2.5}$. As such, this section does not exhaustively review the science updates made in CMAQ53. Additional details on the updates presented here, along with information regarding all the updates made to the model can be found in the CMAQ53 release notes and documentation on the CMAQ GitHub webpage (https://github.com/USEPA/CMAQ).

### 2.1 Chemistry

There are three basic families of chemical mechanisms implemented in CMAQ: Carbon-Bond (CB), Statewide Air Pollution Research Center (SAPRC), and the Regional Atmospheric Chemical Mechanism (RACM). Of the three families of mechanisms available, CB tends to be the most widely used for regional air quality applications with CMAQ, and CB6r3 is the most recent version of CB implemented in CMAQ53. While the CB6r3 name is retained in CMAQ53, the version of CB6r3 in CMAQ53 differs from the original version implemented outside of CMAQ. This section describes updates made in the CB6r3 chemical mechanism and associated aerosol chemistry in CMAQ53.



### 2.1.1 Updates to the chlorine chemistry

The chlorine chemistry in the CB6r3 chemical mechanism (Yarwood et al., 2014; Luecken, et al., 2019) has been updated in CMAQ53. A new model species, chlorine nitrate ($ClNO_3$) and several new reactions (including heterogeneous hydrolysis of $ClNO_3$) have been added. The chlorine chemistry in CMAQ52 contained 26 chemical

reactions and seven chlorine species while the chlorine chemistry in CMAQ53 contains 31 chemical reactions and eight chlorine species. All versions of the CB6 chemical mechanism in CMAQ53 use the same chlorine chemistry. An updated Euler Backward Iterative (EBI) solver was also developed for the revised mechanisms. Model sensitivity simulations show that including the $ClNO_3$ chemistry decreases monthly mean $O_3$ by $0.1 - 0.5$ parts per billion by volume (ppbv). The impact of the updated chlorine chemistry on aerosols was shown to be small.

### 2.1.2 Addition of the bromine/iodine chemistry

Detailed bromine/iodine (halogen) chemistry (Sarwar et al., 2015) was included in a prior version of the CB mechanism (i.e. CB05) in CMAQ52. Sarwar et al. (2019) updated the detailed bromine/iodine chemistry in CB05 and examined its impact on $O_3$ using the hemispheric CMAQ (HCMAQ) model. The bromine/iodine chemistry was then adapted for the CB6r3 chemical mechanism and implemented into CMAQ53. The updated chemistry contains 38 gas-

phase reactions, four heterogeneous and five aqueous-phase reactions for bromine species, and 44 gas-phase and 10 heterogeneous reactions for iodine species. The detailed bromine/iodine chemistry is more active in the HCMAQ model where marine environments represent large extents of the modeled domain in which halogen chemistry influences air masses as they traverse vast extents of oceans during intercontinental transport. However, this update can also be used in the regional version of the model, and there are notable changes along oceanic coastlines. Based

on HCMAQ model sensitivity simulations for October – December 2015, the detailed bromine/iodine chemistry in CB6r3 reduces $O_3$ by an average of $3.0 - 14$ ppbv over much of the open ocean area for the three-month period. Its impact over coastal regions is greater than the interior portion of land, where changes are smaller but non-negligible. The details of the updated bromine/iodine chemistry in CMAQ and its impact on $O_3$ are described in Sarwar et al. (2019).

### 2.1.3 Addition of the DMS chemistry

Dimethyl sulfide (DMS) chemistry has been combined with CB6r3 and implemented into CMAQ53. The DMS chemistry in CMAQ contains seven gas-phase reactions involving DMS and oxidants (OH, $NO_3$, Cl, ClO, IO, BrO). The reactions of DMS with oxidants produce sulfur dioxide ($SO_2$) which can further be converted into sulfate ($SO_4^{2-}$) through both gas and aqueous phase $SO_2$ oxidation pathways already in CMAQ (Sarwar et al., 2011). The combined

chemical mechanism containing CB6r3, detailed bromine/iodine, and DMS chemistry is named CB6r3m. Based on HCMAQ model sensitivity simulations for October – December 2015, introducing DMS chemistry into CB6r3 increases $SO_2$ by an average of $20 - 160$ pptv and $SO_4^{2-}$ by $0.1 - 0.8$ $\mu g\ m^{-3}$ over much of the open ocean area for the three-month period. Changes over land are smaller and primarily limited to coastal regions. The details of the DMS chemistry in CMAQ and its impact are described in Zhao et al. (2020).



### 2.1.4 Updates to the simple halogen mediated $O_3$ loss

Computational efficiency is important for chemical transport models (CTMs) because their use in air quality management requires exploring various emission scenarios. The detailed bromine/iodine (halogen) chemistry increases the computational demand of the model; thus, a simplified first-order rate constant was previously developed for calculating halogen-mediated $O_3$ loss over seawater (Sarwar et al., 2015) for use in regional CMAQ model applications with limited seawater in the domain and incorporated into a previous version of CMAQ. This rate constant has been updated in CB6r3 and incorporated into CMAQ53. The newly calculated first-order surface $O_3$ loss rate is approximately 10% greater than that used in CMAQ521. Model sensitivity simulations over the conterminous U.S. domain with the existing and updated simple first-order $O_3$ loss rate suggests that the updated simple first-order halogen-mediated $O_3$ loss further reduces monthly average $O_3$ by up to 4.0 ppbv over seawater and up to 2.0 ppbv in coastal areas. This simple halogen-mediated $O_3$ loss is used in all chemical mechanisms in CMAQ except in the detailed bromine/iodine chemistry.

### 2.1.5 AERO7/7i

The aerosol chemistry in CMAQ53 has been significantly updated and is available as a new aerosol module called AERO7. The aerosol model from CMAQ521, AERO6, is still available for use with CMAQ53. The version of the aerosol module used by CMAQ is a build-time option that is tailored for the chosen chemical mechanism (e.g. cb6r3_aero6 and cb6r3_aero7), with only that specific aerosol version then available for that specific executable.

The AERO7 module in CMAQ53 improves consistency in representing SOA formation pathways between the CB and SAPRC-based chemical mechanisms; updates monoterpene SOA yields from photooxidation (OH and $O_3$); adds uptake of water onto hydrophilic organics (Pye et al., 2017); adds consumption of inorganic $SO_4^{2-}$ when isoprene epoxydiol (IEPOX) organosulfates are formed; and improves computational efficiency by parameterizing anthropogenic SOA yields through a volatility basis set (VBS) instead of an Odum 2-product fit (Qin et al., 2020). The 21 new species in the AERO7/7i modules are listed in Table S1. In addition, monoterpene nitrates and their SOA products (Pye et al., 2015) are new to the AERO7 module, but were introduced in the AERO6i module in CMAQ521. There were also 28 species that were deprecated in AERO7/7i (see Table S1).

There are several notable differences between the AERO7 and AERO7i modules, primarily related to the degree of speciation of isoprene SOA in each module. Specifically, IEPOX SOA (Pye et al., 2013, 2017), including 2-methyltetrols and organosulfates, is represented using species AISO3J in AERO7. In AERO7i, AISO3J is approximately zero and IEPOX SOA is represented explicitly as organosulfates (AIEOSJ), 2-methyltetrols (AIETETJ), and dimers (AIDIMJ). In addition, AERO7i includes explicit methylglyceric acid (AIMGAJ) and its analogous organosulfate (AIMOSJ), both of which are minor. AERO7i also includes a particle-phase isoprene dinitrate (Pye et al., 2015) that was not ported to AERO7. CMAQ users who require additional isoprene SOA speciation (e.g., to evaluate against measurements) may want to use the AERO7i module. Since AERO7i is only available for use with SAPRC07-based gas-phase chemistry with expanded isoprene intermediates (i.e. SAPRC07tic),


AERO7 (currently only linked with CB-based mechanisms) and AERO7i also result in different SOA abundances due to differences in the oxidant budget driving SOA formation.

Monoterpene oxidation, which accounts for half of the organic aerosol in the southeastern U.S. in summer, was significantly underestimated in CMAQ52 with AERO6 (Zhang et al., 2018) and has been updated in AERO7/7i. The Odum 2-product monoterpene SOA (Carlton et al., 2010) in AERO6 was replaced in AERO7/7i with updated yields based on more recent experimental data by Saha and Grieshop (2016). The new yields are represented using a VBS fit and are applied to both OH and $O_3$ oxidation of monoterpenes. The fit allows for prompt formation of low-volatility material, which is more consistent with recent observations indicating autoxidation is a major contributor to monoterpene SOA (Pye et al., 2019). No additional chemistry, such as oligomerization, is applied to the prompt yields. The updated monoterpene photooxidation matched well with ambient observations in the southeastern U.S. for different seasons (Xu et al., 2018). While seven VBS bins were used in the implementation of Xu et al., the highest volatility bin was not included in the AERO7 implementation as it had very minor contributions to the SOA, even under cold conditions with high loadings.

Finally, there are some emission speciation updates required when using the AERO7 module with CB6r3 in CMAQ53, particularly when using emissions previously generated for the AERO6 module with CB chemistry in CMAQ521. AERO7/7i require that α-pinene (usually denoted APIN) is separate from all other monoterpenes (TERP) in the model. This is to avoid making SOA from α-pinene + nitrate radical reactions, as that pathway has been shown to produce negligible SOA (Fry et al., 2014). All SAPRC07-based mechanisms, including the AERO6/6i ones, already treat APIN as separate and mutually exclusive of TERP and therefore SAPRC07-based AERO6/6i/7/7i emissions will work with any AERO module. CB6r3 with AERO6 continues to include APIN in TERP as it did in CMAQ521.

There are three options available to use CB6r3-AERO6 emissions with the CB6r3-AERO7 module. The first approach is to comprehensively and properly map anthropogenic monoterpene emissions (which are currently relatively minor in the emission inventory) by removing APIN from all other monoterpenes. This approach may be unnecessary when biogenic emissions are calculated inline within CMAQ. The second approach is to use an approximation that assigns 30% of all emitted terpenes to APIN and the remaining 70% to TERP based on the assumption that 30% of global monoterpene emissions are α-pinene (Pye et al. 2010). This approach could be used when biogenic emissions are pre-processed as input files. The third approach is to allow CMAQ to determine the correct biogenic mapping when using inline calculated biogenic emissions. This approach uses separate biogenic emission mapping profiles for CB6r3-AERO6 and CB6r3-AERO7 available within CMAQ and is not an option when biogenic emissions that were processed off-line (pre-processed) for AERO6 and are used for AERO7.

While more details regarding the impact of the new AERO7 module on model performance will be presented in section 4 on model evaluation, the overall impact on model performance from the updates in AERO7 is to increase $PM_{2.5}$ mass, primarily in summer, in vegetation rich locations such as the southeast U.S. (Xu et al., 2018). Ambient $PM_{2.5}$ is further increased by water uptake, which modulates aerosol optical properties and has implications for metrics such as aerosol optical depth (AOD) that represent *in situ* (vs. dry) conditions.



### 2.1.6 Other aerosol processes

Several other more minor updates were made to the aerosol processes in CMAQ53. Specifically, the dry deposition velocities, particularly for coarse-mode particles, were too high by 10-100% in CMAQ521. Detailed testing revealed that the dry deposition algorithm in CMAQ521 was not suitable for coarse-mode particles, especially when the mode width (σ) of the coarse mode approached 2.5, the upper bound. The revised algorithm in CMAQ53 reduces the strong dependence on σ and introduces a dependence on leaf area index (LAI). The LAI dependence is intended to capture

the larger deposition hypothesized over, for example, forest canopies, relative to bluff-body surfaces. The net result of this algorithm update is a large reduction in coarse-mode particle deposition in areas of high σ. However, both Aitken- and coarse-mode particle deposition increase in highly vegetated areas, such as the southeast U.S.

The gravitational settling algorithm in CMAQ53 was also updated because it did not conserve mass for certain aerosol size distribution parameter combinations in earlier version of CMAQ. These errors typically occurred in the highest-

altitude model layers but could propagate to other parts of the domain and lower levels if the errors were large enough. CMAQ53 resolves these problems by correcting a minor error in the time-step increment and by maximizing the number of iterations used to calculate settling. While these updates resolve the mass accumulation errors (up to $10 - 50\%$ of the total coarse mode particle mass), they minimally impact overall model performance for $PM_{2.5}$ and $PM_{10}$ given the highly infrequent and transient nature of the errors.

### 2.1.7 Aqueous and heterogenous chemistry

AQCHEM-KMT was introduced in CMAQv51 to provide a framework for examining detailed cloud chemistry and increase connections between cloud microphysics and chemistry by introducing a dependence of mass transfer and chemistry on droplet size (Fahey et al., 2017). KMT refers to the category of cloud chemistry modules that are built using the Kinetic Pre-Processor (KPP) version 2.2.3 (Damian et al., 2002) and treat the mass transfer between gas and

aqueous phases with the resistance model of Schwartz (1986). The use of KPP facilitates the straight-forward extension of the mechanism to include treatment of additional aqueous chemical reactions beyond those considered in CMAQ's default cloud chemistry module (AQCHEM). Two additional KMT cloud chemistry options are available in CMAQ53: KMT version 2 (KMT2) (Fahey et al.,2017; Fahey et al., 2019) and KMTBR (Sarwar et al., 2019). KMTBR expands the default cloud chemistry mechanism of five S(IV) oxidation and two SOA reactions to include

the treatment of aqueous chemistry for several bromine species. KMT2 builds upon CMAQ's existing in-cloud sulfur (S) oxidation chemistry, replacing the yield parameterization of in-cloud SOA formation from GLY/MGLY+OH with a mechanistic representation of small dicarboxylic acid formation from the reactions of OH with glyoxal, methylglyoxal, glycolaldehyde, and acetic acid (Lim et al., 2005; Lim et al., 2010, 2013; Sareen et al., 2013). It also includes additional aqueous chemistry for S, nitrogen (N), carbon (C), and oxygen-hydrogen ($OH^-$) species largely

based on the aqueous mechanism of ReLACS-AQ (Leriche et al., 2013), a compact cloud-chemistry mechanism built from and tested against more comprehensive models.

The species impacted and magnitude of the effects from the KMT2 update depend heavily on season and location. Simulations over the conterminous U.S. for July 2016 showed a surface-level increase in in-cloud SOA from small



carbonyl compounds (AORGC) of 400 – 600% compared to CMAQ's standard yield-based parameterization for GLY
and MGLY+OH, increasing July average organic aerosol by up to 0.5 µg m$^{-3}$ in areas of the southeastern U.S. KMT2
also shows elevated cloud SOA at the surface and aloft. A recent HCMAQ simulation using KMT2 showed monthly
average changes to in-cloud SOA of up to 1.0 µg m$^{-3}$ in regions and during seasons when oxidant levels and biogenic
emissions were high, increasing by at least a factor of five in much of the southeast U.S. during the summer.

In the winter, $SO_4^{2-}$ is the most significantly impacted species from this update. HCMAQ simulations for January 2016
indicate a monthly average increase in $SO_4^{2-}$ concentrations of up to 17% over China and 34% over the U.S. Nitrate
decreased with a similar pattern to the $SO_4^{2-}$ increase. Monthly average $O_3$, HCHO, and $NO_x$ were minimally impacted
over the conterminous U.S. (typically within approximately 3%), but larger absolute impacts over shorter time periods
and in other regions (e.g., change in average $O_3$ up to -11% over Asia) were observed. Impacts of KMT2 on CMAQ
runtime depend heavily on domain, season, and chemical mechanism, but it can increase runtime by 30% or more for
some CB6-based HCMAQ applications as the number of cloudy grid cells increases. As such, KMT2 is considered a
research option; and while not used in the regional CMAQ simulations presented here, a special version of KMT
(KMTBR) is used in the HCMAQ simulations.

### 2.2 Air-surface exchange

CMAQ53 offers two air-surface exchange models: updated M3Dry and the new Surface Tiled Aerosol and Gaseous
Exchange (STAGE) model. For dry deposition, both options are modified versions of the previous M3Dry model.
However, $NH_3$ bi-directional exchange (BiDi) in both options diverges from the previous M3Dry model. In both
M3Dry and STAGE, the cuticular resistance of non-ionic organic species is modeled similarly to the processes
involving the partitioning of semi-volatile gases to organic aerosols. Bulk leaf wax properties and composition are
taken from the observations of Schreiber and Schoenherr (2009), and vapor pressure is used to estimate air cuticle
partitioning following Raoult's law. Values are then used to populate a "relative reactivity" field in CMAQ, but the
cuticular resistance is essentially converted from a reactivity-driven process to an absorptive partitioning process (for
organic species). The following sections describe the specific updates to M3Dry in CMAQ53 and provide a brief
description of the new STAGE model. More details on each model can be found in their respective publications
referenced in each section.

### 2.2.1 Updates to the M3Dry deposition model

The new M3Dry BiDi option requires fertilization inputs that are generated by the Environmental Policy Integrated
Climate (EPIC; Williams, 1995) model through the Java-based Fertilization Emission Scenario Tool for CMAQ
(FEST-C; Ran et al., 2019) interface. M3Dry in CMAQ53 has been updated to directly use the information provided
by the EPIC model. M3Dry uses daily input of total ammonium ($NH_4^+$) from EPIC in the 1- and 5-cm soil layers, soil
pH, soil moisture content, soil cation exchange capacity (CEC), and soil texture parameters for each of 42 crop types.
These data are read into CMAQ and used to compute the amount of $NH_4^+$ available for volatilization in the soil and
the soil $NH_3$ compensation concentration. Bi-directional $NH_3$ flux is computed from the soil compensation
concentration (the maximum of the 1- or 5-cm layer) according to the resistance model described in Pleim et al. (2013).





Hourly net bi-directional $NH_3$ flux, $NH_3$ dry deposition, and $NH_3$ emissions are all output in the CMAQ DRYDEP
file. The capability to output land-use specific dry deposition flux has been removed from M3Dry but can be estimated
off-line using available post-processing tools, which is particularly useful for ecosystem applications.

The new $NH_3$ bi-directional flux model in M3Dry substantially changes $NH_3$ concentrations and surface flux which
depend on EPIC input data. The model is identical to the box model developed and evaluated against field flux
measurements in Pleim et al. (2013). Details of the implementation in CMAQ53 and evaluation compared to ground
sites and satellite retrievals are presented by Pleim et al. (2019). Unlike previous versions of M3Dry, the new scheme
relies on direct coupling to EPIC, which includes comprehensive C-N-P cycles and daily agricultural management, to
better model the soil biogeochemistry related to the soil $NH_4$ and its availability to volatilize. Additional information
on the impact on model performance of the updates to the M3Dry scheme in CMAQ53 are provided in Sect 4 on
model evaluation.

Several other important updates in M3Dry related to air-surface exchange depend on land-use and areas with snow
cover. First, the dry deposition to snow was updated in M3Dry to increase the $O_3$ dry deposition resistance to snow.
Specifically, the resistance was increased from 1000 to 10000 s m$^{-1}$ based on observations from Helmig et al. (2007)
so that the deposition velocity ($V_d$) of $O_3$ to snow is on the order of 0.01 cm s$^{-1}$. The result is a significant increase in
ambient $O_3$ mixing ratios over snow-covered areas in CMAQ53 with M3Dry. The ground resistance ($R_g$) for $O_3$ has
been changed to be sensitive to soil moisture ($W_g$) following Meszaros et al. (2009) and Fares et al (2012):

$$R_g(O_3) = 200 + 300\ W_g/W_{fc}, \hspace{4cm} (1)$$

Where $W_{fc}$ is the soil moisture content at field capacity and $R_g(O_3)$ is limited to 500 s m$^{-1}$ when $W_g \geq W_{fc}$. The result
is a decrease in the $R_g(O_3)$ for soil compared to CMAQ521 where $R_g(O_3) = 667$ s m$^{-1}$, which decreases ambient $O_3$
mixing ratios, as more $O_3$ is dry deposited to the soil.

**2.2.2 Surface Tiled Aerosol and Gaseous Exchange (STAGE) deposition model**

With CMAQ53, a new deposition model, STAGE, is available. The STAGE deposition model estimates fluxes from
sub-grid cell fractional land-use values, aggregates the fluxes to the model grid cell, and unifies the bi-directional and
uni-directional deposition schemes using the resistance model frameworks of Massad et al. (2010) and Nemitz et al.
(2001). Since STAGE utilizes fractional land use, the land-use specific fluxes and deposition velocities across the
mosaic of land-use categories are output for each grid cell.

Bidirectional $NH_3$ exchange, which utilizes the widely implemented resistance model framework of Nemitz et al.
(2001) with the cuticular resistance from Massad et al. (2010), is integrated into STAGE in CMAQ53 following Bash
et al. (2013). Additionally, the nitrification rates are now estimated from EPIC model output, along with
$NH_4^+$ production from organic nitrogen mineralization. Soil resistance formulation is based on maximum diffusive
depth following Kondo et al. (1990) and Swenson and Lawrence (2014). Soil moisture at plow depth, 5 cm, is
estimated from gravitational draining in STAGE rather than the weighted mean of the 0.01 and 1.0 m soil layers in



M3Dry. The previous implementation of the bi-directional mercury (Hg) algorithm in CMAQ was simplified and modified to provide soil, vegetation, and water Hg concentrations, and to integrate the fluxes from the STAGE model.

The in-canopy aerodynamic resistance in STAGE is derived by integrating the in-canopy eddy diffusivity as parameterized by Yi (2008) from zero to the full canopy LAI. The parameterization of deposition to snow was also changed from the original M3Dry implementation to assume that, snow, when present, covers vegetation and soil, which eliminates the deposition pathways to those surfaces. Ozone deposition to soil in STAGE is based on under-canopy measurements from Fares et al. (2014) and Fumagalli et al. (2016), similar to the revised M3Dry deposition parameterization discussed in Sect. 2.2.1. The resistance to deposition to wet terrestrial surfaces was modified following Fahey et al. (2017) to align with the AQCHEM-KMT2 aqueous parameterization. The most significant change is the resistance parameterization now has a bulk accommodation coefficient that occasionally becomes the limiting resistance for highly soluable compounds, most notably $SO_2$, when the surface is wet and the aerodynamic resistance is low. This results in maximum deposition velocities of ~7.0 cm s$^{-1}$ which agrees well with observed maximum deposition values (Wu et al. 2018). Aerosol deposition in STAGE follows the changes to M3Dry in CMAQ53, but with the following differences. Aerosol impaction for vegetated surfaces is parameterized following Slinn (1982) for vegetated surfaces using the characteristic aerodynamic leaf radius for plant functional types from Zhang et al. (2001), and Giorgi (1986) for soil and water surfaces. Aerosol deposition velocities are then estimated for both smooth and vegetated surfaces and area weighted by the vegetation coverage.

## 2.3 CMAQ emission processing updates

### 2.3.1 Detailed Emission Scaling, Isolation, and Diagnostic (DESID) module

To address specific research questions and isolate emission sources, CMAQ allows users to set and scale emissions on a per-species basis for gases and aerosols, and disable specific inline modules such as the Biogenic Emission Inventory System (BEIS; https://www.epa.gov/air-emissions-modeling/biogenic-emission-inventory-system-beis), wind-blown dust (WBD), and lightning NO. Yet, prior to CMAQ53, users could not run with multiple gridded emission files, scale a species from different sources independently, disable sea-spray emissions, or easily determine how aerosol size distributions were applied.

The Detailed Emission Scaling, Isolation, and Diagnostic (DESID) module is introduced in CMAQ53 to allow the consideration of emissions from external sources, such as those processed using the Sparse Matrix Operator Kernel Emissions (SMOKE), along with modules within CMAQ, such as WBD or those processed using BEIS. Through the emission control file (formatted as a Fortran namelist), DESID allows users to map chemical species from emissions input files to CMAQ model species, zero out or scale emissions from individual offline and inline sources, and introduce complex scaling rules along user-defined boundaries. Specifically, useful features include the following:

- Gases and aerosols can be scaled independently for different sources

- All or a subset of emissions can be scaled with respect to a gridded mask identifying a geographical area (e.g. a state or region)





- Emission rates can be scaled on a unit basis or by conserving moles or mass at the user's discretion

- Families of chemical species, emission sources, or geographical areas can be defined to simplify the user's interactions via the control file and ensure confidence in application of scaling rules

- Diagnostic files may be output for any emission source so that scaling instructions can be quality assured

or used to assess impacts

Mechanism-specific emission control files are provided with the CMAQ repository. DESID also writes an extensive diagnostic output to the CMAQ log file to inform users about species from emission input files that were not used, emissions species that were missing from the input files, and scale factors that were successfully applied for each source, species, and mask. Note that considerations must be made when scaling emissions that are represented in other parts of the modeling system. For example, when scaling $NH_3$ emissions using DESID the $NH_3$ emissions generated

using the EPIC model must also be scaled accordingly when using BiDi.

### 2.3.2 BEIS chemical mapping update

Two minor updates were made to BEIS input and diagnostic output in CMAQ53. First, a lookup table was introduced that maps CMAQ chemical mechanisms to the mechanism field on the BEIS speciation profile (GSPRO) file.

Previously, users selected mechanisms on the GSPRO file (distributed as part of the CMAQ code repository) to be compatible with the CMAQ chemical mechanism selected. This update protects users from incorrectly matching CMAQ chemical mechanisms to mechanism names in the GSPRO input file. Second, for diagnostic output, an error was corrected in the initialization of a variable used to accumulate emissions estimated during sub-time steps in CMAQ521 which had previously resulted in an overestimate of BEIS diagnostic emissions, as the running sum was

accumulating too much mass in successive output time steps. Note that this error in the diagnostic output, which has been corrected in CMAQ53, did not affect the emissions computed by BEIS that were used within prior versions of the model.

### 2.4 Meteorology processing updates

The Meteorology-Chemistry Interface Processor (MCIP; Otte and Pleim, 2010) was updated to versions 5.0 (MCIP50)

and 5.1 (MCIP51) concurrently with the releases of CMAQ53 and CMAQ531, respectively. MCIP50 includes several major changes and corrects some minor coding errors. Of note, support for the hybrid vertical coordinate (HVC) system in the Weather Research and Forecasting model (WRF; Skamarock and Klemp, 2008) was introduced in MCIP50 through adjustments to the Jacobian. Consequently, layer collapsing was removed from MCIP, in part, because of the complexities of developing an algorithm that is also compatible with HVC. Further, support for

MM5—the predecessor to WRF—was discontinued. Lastly, the output routines were overhauled to accommodate a new option to create MCIP output in network common data form (netCDF). MCIP51 consisted of two minor corrections and one minor extension to MCIP50.

### 3 CMAQ model inputs, configuration, and data sets

CMAQ model simulations require numerous input files. The most important and complex input files are meteorology
and emissions. Like most other offline air quality models, CMAQ does not compute its own meteorology or
anthropogenic emission inputs (although it can compute natural emissions), but instead relies on separate models to
compute those inputs. CMAQ also requires initial conditions (ICs) and lateral boundary conditions (BCs) for chemical
species. The following sections describe generating the input meteorology, emissions, and BCs used in the CMAQ
simulations performed here, followed by sections describing the CMAQ model configuration options and observation
data sets used to evaluate model performance.

### 3.1 WRF meteorology inputs

For CMAQ, the primary meteorology model supported and used in the community is WRF. For the results presented
here, two recent versions of the WRF model were used, 3.8 (WRF38) and 4.1.1 (WRF411). The CMAQ simulations
used to evaluate the impact of the science updates in CMAQ53 and impact of the BCs, presented in Sect. 4.1 and 4.2
respectively, utilize meteorology from a WRF38 simulation, while the results utilizing the WRF411 meteorology are
presented as their own sensitivity in Sect. 4.3. Both the WRF38 and WRF411 simulations used the same scientific
options, and it is noted where either the options employed in WRF differ or where the model updates between WRF
versions have a significant effect on the meteorology results.

The WRF options common to both simulations include the Rapid Radiation Transfer Model Global (RRTMG) for
long- and short- wave radiation (Iacono et al., 2008), the Morrison microphysics scheme (Morrison et al., 2005), and
the Kain-Fritsch (KF) cumulus parametrization scheme (Kain, 2004). The Pleim-Xiu land-surface model (PX-LSM;
Pleim and Gilliam, 2009) and Asymmetric Convective Mixing 2 planetary boundary layer (PBL) model (ACM2;
Pleim, 2007a,b) were used along with the Pleim surface layer scheme (Pleim, 2006). Four-dimensional data
assimilation (FDDA) via analysis nudging was also employed in the WRF simulations. Grid and soil moisture nudging
data were both provided by the North America Model (NAM) reanalysis data set (Mesinger et al., 2006). The 40-
category "NLCD40" land-use data set, which uses National Land Cover Data classifications for the U.S. and Moderate
Resolution Imaging Spectroradiometer (MODIS) satellite derived land-use elsewhere, was used and defined
parameters such as surface roughness and albedo in the PX-LSM. However, note that for the WRFv411 simulation,
the PX LSM did not use LAI and areal fraction covered by vegetation (VF) based on the NLCD40 categories but
instead directly used MODIS satellite-derived information (described in more detail below). Both WRF simulations
used lightning assimilation (Heath et al., 2016) to improve the location and intensity of precipitation in the model.

There were many updates to the WRF system from version 3.8 to version 4.1.1 but only a few that are relevant to
these CMAQ applications. Notably, the HVC system is used in the WRF411 simulation, whereas the conventional
sigma vertical coordinate (SVC) system is used in the WRF38 simulation. In HVC, the vertical levels follow the
terrain near the surface and use isobaric surfaces aloft. HVC can reduce the artificial influence of topography towards
the top of the model, which can lead to spurious vertical motions with SVC, particularly in complex terrain. Both
CMAQ and the MCIP were updated to support the HVC. The other noteworthy differences between WRF versions
used here are in the PX-LSM. The most important of the changes to the PX-LSM were the specification of the
vegetation related parameters: LAI and VF. In WRF38 and earlier versions, LAI and VF were specified from a look-





up table where each land-use category was assigned values for minimum and maximum LAI and VF. The grid cell values are averaged by fractional land-use coverage with a seasonality function using deep-soil temperature to interpolate between maximum and minimum values.

For WRF411 there are two options available for vegetation parameters: an updated look-up table and direct input of data from MODIS satellite products. The WRF411 simulation used in this study used the MODIS derived LAI and
fraction of absorbed photosynthetically active radiation (FPAR) (used as surrogate for VF) as described by Ran et al. (2016). The biggest impact of this difference between WRF simulations is that the VF is much smaller in the sparsely vegetated parts of the domain, such as most of the western part of North America (Fig. S1). Note that the new version of the vegetation parameter look-up table starting in WRF version 4.0 was developed to complement the MODIS parameter values. Also, the soil parameter calculations in the PX-LSM were modified to use analytical functions from
Noilhan and Mahfouf (1996) for field capacity, saturation, and wilting point based on fractional soil data. The namelists used for each WRF simulation are provided in the supplemental material (Tables S2 and S3). Model ready meteorological input files were created using MCIP version 4.3 for WRF38 data and version 5.0 for WRF411 data.

### 3.2 Emissions

### 3.2.1 Regional emission inputs

Emission inputs for the 2016 CMAQ regional (i.e. conterminous U.S. at 12km horizontal grid spacing) simulations were developed through a collaboration between EPA, States, and multi-jurisdictional planning organizations (MJOs), referred to as the "emission modeling platform" (EMP; https://www.epa.gov/air-emissions-modeling/2014-2016-version-7-air-emissions-modeling-platforms). The 2016 EMP went through several iterations before being designated as final. The 2016 version 1 EMP used here is described at http://views.cira.colostate.edu/wiki/wiki/10202. The major
emission components of the 2016 EMP are summarized below.

The 2016 EMP is built upon the EPA's 2014 NEI version 2 (2014NEIv2). The 2014NEIv2 includes five data categories: point, non-point, non-road mobile, on-road mobile, and events consisting of fires (e.g. prescribed and wildland fires). The NEI uses 60 sectors to delineate the emissions. Emissions from the Canadian and Mexican inventories and several other non-NEI data sources are included in the 2016 EMP. The point source emission
inventories include partially updated emissions for 2016 using Continuous Emission Monitoring System (CEMS) values for $NO_x$ and $SO_2$ when available. Agricultural and wildland fire emissions are specific to 2016. Most area source sectors use 2014NEIv2 emissions estimates except for commercial marine vehicles (CMV), fertilizer emissions, oil and gas emissions, and on-road and non-road mobile source emissions. For CMV, $SO_2$ emissions were updated to reflect new regulations on sulfur emissions that took effect in 2015. For fertilizer $NH_3$ emissions, a 2016-
specific emission inventory was used. Area source oil and gas emissions were projected from 2014NEIv2 to better represent 2016. On-road and non-road mobile source emissions were developed using the Motor Vehicle Emission Simulator version 2014a (MOVES2014a; https://www.epa.gov/moves). On-road emissions were developed based on emissions factors output from MOVES2014a for the 2016 run with inputs derived from the 2014NEIv2 including activity data projected to 2016.




### 3.2.2 Hemispheric emission inputs

Emissions for the HCMAQ simulations over the Northern Hemisphere at 108-km horizontal grid spacing follow the 2010 Hemispheric Transport for Air Pollution version 2 (HTAPv2) (Janssens-Maenhout et al., 2015) with updates to regional emission inventories (e.g., North American, China). The model-ready emissions were developed by Vukovich et al. (2018), with the anthropogenic portion derived from the HTAPv2 inventory and the natural portion using Model of Emissions of Gases and Aerosols from Nature (MEGAN; Guenther et al., 2012), Global Emissions IniAtive (GEIA; http://www.geiacenter.org), and Fire INventory the from National Center for Atmospheric Research (FINN; Wiedinmyer et al., 2011) to represent biogenic VOC, soil NO, lightning NO, and biomass burning. The HTAP emissions are on a 0.1° grid and include CO, non-methane VOC, $NO_x$, $SO_2$, $NH_3$, $PM_{10}$, $PM_{2.5}$, black carbon, and OC from agriculture, aircraft, industry, energy production, ground transportation, residential and shipping.

### 3.3 Boundary conditions

Several different sets of BC inputs are used and evaluated for the regional CMAQ simulations presented here. All are derived from 108-km HCMAQ simulations utilizing 44 vertical layers on a polar stereographic grid. These BCs from each HCMAQ simulation correspond to the version of CMAQ used for the regional-scale simulation, thereby representing a complete system update from CMAQ521 to CMAQ53. Specifically, the CMAQ521 simulation uses BCs from a corresponding HCMAQ521 simulation utilizing WRF38 meteorology, while the CMAQ53 simulations utilize BCs from HCMAQ53 simulations also using WRF38 that is then processed corresponding to the vertical coordinate used in the regional CMAQ simulation. Although the CMAQ code used in the HCMAQ simulations is identical to code used in regional CMAQ simulations, there are several important configuration differences to note.

First, the HCMAQ simulations use the computationally intense implementation of the DMS/halogen chemistry to more comprehensively reflect chemistry over open ocean area, while the regional domain uses the simplified halogen chemistry. Second, a potential vorticity (PV) scaling technique (Xing et al., 2016; Mathur et al, 2017) is applied in the HCMAQ simulations to more accurately represent $O_3$ in the top layer of the model and represent the impacts of stratosphere-troposphere exchange on tropospheric $O_3$. The HCMAQ simulations also have WBD enabled, but do not implement the empirical potential SOA from combustion sources (PCSOA) or semi-volatile primary OA (SVPOA) options (Murphy et al., 2017).

The HCMAQ simulations use the CB6R3M_AE7_KMTBR chemical mechanism while the regional CMAQ use the CB6r3_AE7_AQ chemical mechanism. The dry deposition scheme used is consistent in paired hemispheric and regional simulations. WRF38 is used to drive the HCMAQ521 and HCMAQ53 simulations; however, the BC files are reprocessed for the CMAQ531 regional simulations that use WRFv411 meteorology to account for the HVC. The hemispheric WRF simulations cover the same horizontal and vertical layers as the HCMAQ simulations. The HCMAQ options and inputs are listed in Table 1.





### 3.4 CMAQ Configuration

All the regional-scale simulations performed here utilize a domain covering the conterminous U.S., northern Mexico, a large portion of southern Canada, and the eastern Pacific and western Atlantic oceans. The domain consists of 299
south-north by 459 west-east grid cells utilizing 12-km horizontal grid spacing and 35 vertical layers with varying thickness from the surface to 50 hPa on a Lambert-Conformal projection. The mid-point of the lowest layer is approximately 10 m above ground level. The simulation period covers 2016, with a 10-day simulation spin-up period from 22-31 December 2015 to minimize the effect of ICs. To further reduce the effects of ICs, 22 December 2015 is initialized using chemical values derived from an older 2015 CMAQ simulation, representing authentic (rather than
climatological) initial values. The regional simulations employed both SVPOA and PCSOA to improve estimates of PM emissions. PCSOA relies on a precursor VOC emission scaled to emissions of POA to form SOA. The parameters developed to simulate PCSOA were constrained with data that represent the qualities of urban, motor-vehicle dominated locations. Thus, PCSOA is most reliably used when wood burning sources of POA, i.e. residential wood combustion (RWC) and wildland fires, are distinguishable in the emission inputs and not included in the scaling of
the precursor VOC to POA emissions. The WBD option was not used in the regional CMAQ simulations to avoid generating anomalously high WBD concentrations under certain conditions. The simulations utilize either the M3Dry deposition model with or without BiDi enabled or the STAGE model with BiDi enabled (Table 1). Other options are common to all the simulations, including the EBI chemical solver, the piecewise parabolic method (PPM) for horizontal and vertical advection, minimum eddy diffusivity ($K_z$), ocean halogen chemistry and sea-spray aerosol
emissions, surface HONO interaction, gravitational settling, inline biogenic (BEISv3.61) emissions, inline point emission plume rise, and version 5 of the Biogenic Emissions Landcover Database (BELD5) created using the Spatial Allocator (SA) Raster Tools system (http://www.cmascenter.org/sa-tools/). All the CMAQ531 simulations use the AERO7 module (CB6r3_AE7_AQ), while the CMAQ521 simulation utilized the AERO6 module (CB6r3_AE6_AQ). Lightning NO emissions were calculated inline using National Lightning Detection Network (NLDN) data to improve
the representation of tropospheric NO mixing ratios (Kang et al., 2019a,b). Although CMAQ531 is used for all the regional-scale simulations, the model performance is unchanged from CMAQ53, as the updates in CMAQ531 addressed only minor errors in CMAQ53 that did not affect the options used here.

### 3.5 Air quality observations

Model estimated concentrations are compared to ambient measurements from several North American air quality
observation networks. Measurements of $O_3$ and $PM_{2.5}$ are acquired from the USEPA's Air Quality System (AQS; https://www.epa.gov/aqs), which culls data from networks maintained by various federal, State, and tribal agencies (e.g., USEPA, National Parks Service). The AQS is the primary source of $O_3$ and $PM_{2.5}$ measurements used here to assess the CMAQ model performance. For 2016, the AQS includes 1,304 hourly $O_3$ monitors (although many sites are inactive during cooler months) and 2,010 hourly/daily $PM_{2.5}$ monitors for the conterminous U.S. with valid data.
Measurements of $O_3$ and $PM_{2.5}$ for Canada are obtained from the National Air Pollution Surveillance (NAPS; https://www.canada.ca/en/environment-climate-change/services/air-pollution/monitoring-networks-data/national-air-pollution-program.html) network, consisting of 190 $O_3$ monitors and 196 $PM_{2.5}$ monitors in 2016 with valid data.





Observations of speciated PM$_{2.5}$ components (e.g. SO$_4^{2-}$, NO$_3^-$, OC) for the U.S. are obtained from the USEPA's Chemical Speciation Network (CSN; https://www.epa.gov/amtic/chemical-speciation-network-csn), the Interagency

Monitoring of PROtected Visual Environments (IMPROVE; http://vista.cira.colostate.edu/Improve/) network, and the Clean Air Status and Trends Network (CASTNet; https://www.epa.gov/castnet). For 2016, valid data were reported for 242 distinct CSN sites (not all species are measured at all sites), located in primarily urban environments; 149 IMPROVE sites, located in primarily rural environments and national parks; and 94 CASTNet sites, located primarily in the eastern U.S. Finally, the Ammonia Monitoring Network (AMON; http://nadp.slh.wisc.edu/amon/), part of the

U.S. National Acid Deposition Program (NADP), provides measurements of ground level NH$_3$ across approximately 90 sites for 2016 and is used here quantify differences in simulations that used M3Dry and STAGE deposition models.

Model values are paired in space and time to observed values using the Atmospheric Model Evaluation Tool (AMET; Appel et al., 2011) version 1.4 (https://github.com/USEPA/AMET). It is worth noting that representativeness (incommensurability) issues are present whenever observed data for a particular point in space and time are compared

to gridded values from a deterministic model such as CMAQ, as deterministic models calculate the average outcome over a grid for a certain set of given conditions, while the stochastic component (e.g. sub-grid variations) embedded within the observations cannot be accounted for in the model (Swall and Foley, 2009). These issues are reduced somewhat for networks that observe for longer durations, such as the daily average values from the CSN and IMPROVE networks and weekly average values from CASTNET, as the longer temporal averaging helps reduce the

impact of stochastic processes. To quantitively assess model performance, several statistical values are calculated and presented in Sect. 4 and the supplemental material. These values are mean bias (MB), normalized mean bias (NMB), root mean square error (RMSE), and the Pearson correlation coefficient (r or COR). The definitions for these metrics are:

$$MB = \frac{1}{N}\sum_1^N (C_M - C_O) \qquad (2)$$

$$NMB = \frac{\sum_1^N (C_M - C_O)}{\sum_1^N C_O} \times 100\% \qquad (3)$$

$$RMSE = \sqrt{\frac{1}{N}\sum_1^N (C_M - C_O)^2} \qquad (4)$$

$$r = \frac{\sum_1^N (C_M - \overline{C_M})(C_O - \overline{C_O})}{\sqrt{\sum_1^N (C_M - \overline{C_M})^2 \sum_1^N (C_O - \overline{C_O})^2}} \qquad (5)$$

where $C_M$ and $C_O$ are simulated and observed concentrations, respectively; $\overline{C_M}$ and $\overline{C_O}$ are the simulated and observed mean concentrations, respectively; and N is the total number of individual observations.

**4 Results**





Sect. 2 highlighted the major scientific updates incorporated into CMAQ53. In some cases, these updates can have a significant impact on model performance. In this section, we quantify the impacts of the science updates in CMAQ53 on model performance, particularly ground level $O_3$ and $PM_{2.5}$. In addition to quantifying the impact of science updates in CMAQ53, we also examine the impacts of several specific updates on the CMAQ modeling system performance, since users often update not just the model version, but also the model inputs when transitioning to a new version of CMAQ. This evaluation is accomplished by comparing results from model simulations that not only use different versions of the model (i.e. CMAQ521 and CMAQ531) with all the same inputs, but also comparing model simulations that use the same version of the model (i.e. CMAQ531) but with different inputs (BCs, meteorology) or science options (M3Dry, STAGE). Collectively these updates represent the transition from the CMAQ521 modeling system to the CMAQ531 modeling system by also considering model inputs. The model performance analysis focuses primarily on the criteria pollutants MDA8 $O_3$ and daily average $PM_{2.5}$, but performance of other species (e.g., OC) is examined when relevant. The impact of the science updates in CMAQ531 is examined in Sect. 4.1; the impact of the science updates on the HCMAQ simulations (and hence BCs) is examined in Sect. 4.2; the impact of transitioning from WRF38 (SVC) to WRF411 (HVC) is examined in Sect. 4.3; and a brief comparison of the M3Dry and STAGE deposition models is presented in Sect. 4.4.

### 4.1 Impact of science updates in CMAQ version 5.3

Here we compare results between a CMAQ521 simulation and similarly configured CMAQ531 simulations. All simulations utilize the same 2016 v1 EMP data, WRF38 meteorology, and M3Dry deposition model. To completely represent the science and system updates made in CMAQ531, three CMAQ531 simulations with slightly different configurations are presented in this section. The first simulation uses CMAQ531 without BiDi enabled and with the application of PCSOA to RWC (CMAQ531_WRF38_M3Dry_noBiDi_RWC). This simulation most closely mimics the configuration of the CMAQ521 simulation (specifically no BiDi and PCSOA applied to RWC) and therefore is intended to isolate the impact of just the science updates in CMAQ53. The second CMAQ531 simulation removes the application of PCSOA to RWC using the DESID module with the ability to read multiple gridded emission files in CMAQ531 and represents how PCSOA was intended to be applied in the model (CMAQ531_WRF38_M3Dry_noBidi). The final CMAQ531 simulation in this section builds upon the second simulation by implementing the M3Dry BiDi option in the model (CMAQ531_WRF38_M3Dry_BiDi), highlighting the impact including that process has on model estimated concentrations. There are two additional CMAQ531 simulations using WRF411: one with M3Dry (CMAQ531_WRF411_M3Dry_BiDi) presented in Sect. 4.3 and one with STAGE (CMAQ531_WRF411_STAGE_BiDi) presented in Sect. 4.4. A list of CMAQ simulation inputs and configuration options for these simulations is presented in Table 1.

Figure 1 presents a time series of monthly average MDA8 $O_3$ bias for all AQS sites within the 12-km domain for the CMAQ521 and five CMAQ531 simulations. Similar time series plots for MDA8 $O_3$ mixing ratio, RMSE and COR are in Figs. S2 – S4, along with spatial plots of seasonal MDA8 $O_3$ bias for the CMAQ521 simulation (Fig. S5). Other than the winter months (Dec, Jan, Feb), average MDA8 $O_3$ is consistently lower by approximately 1.0 – 2.0 ppbv in the three CMAQ531 simulations that use WRF38 versus the CMAQ521 simulation, and there is little difference



between the three CMAQ531 simulations, indicating that removing the application of PCSOA on RWC and enabling BiDi, as expected, minimally affect monthly average $O_3$ mixing ratios. While there is considerable improvement in summertime and wintertime bias and RMSE (Fig. S3) in the CMAQ531 simulations, COR (Fig. S4) is slightly higher
throughout the year in the CMAQ521 simulation. The lower $O_3$ mixing ratios outside of winter are primarily due to the update to Rg ($O_3$) in CMAQ531 which decreases the resistance of $O_3$ dry deposition to soil, thereby increasing $O_3$ dry deposition and decreasing ambient $O_3$ mixing ratios. In the winter, the decrease in $O_3$ dry deposition to snow in CMAQ531 results in higher average ambient $O_3$ mixing ratios due to more prevalent snow cover.

This latitudinal effect on $O_3$ mixing ratios from snow cover is apparent in Fig. 2, which presents the difference in
seasonal   average   MDA8   $O_3$   absolute   bias   between   the   CMAQ521   simulation   and   the CMAQ531_WRF38_M3Dry_BiDi simulation. In winter, sites in the U.S. northern latitudes and Canada indicate a seasonal average reduction in MDA8 $O_3$ bias of 10 ppbv or more with CMAQ531. Sites in the lower latitudes show little change in bias, with some sites in the southern U.S. showing a slight increase in bias with CMAQ531. In spring (Mar, Apr, May), MDA8 $O_3$ bias is broadly higher with CMAQ531, with bias increasing from north (where the update
of deposition to snow still results in lower bias) to south, where seasonal average biases are higher by up to 4.0 – 5.0 ppbv.

In summer (Jun, Jul, Aug), bias is lower in the CMAQ531_WRF38_M3Dry_BiDi simulation by 2.0 – 4.0 ppbv across much of the U.S. and Canada, with the largest exception in the southwestern U.S. (including California) where the bias increases by 3.0 – 6.0 ppbv at many sites. Previous studies have shown that mean $O_3$ is often overestimated by
CMAQ in the summer (except in California), so the broad decrease in $O_3$ with CMAQ531 reduces the bias at most sites but worsens underestimation in California. The pattern in fall (Sep, Oct, Nov) is similar to summer, with a large, broad decrease in bias for most sites outside of the southwestern U.S. with CMAQ531. Sites in southern Florida indicate higher bias with CMAQ531 (3.0 – 4.0 ppbv), as do some other sites in the southeast U.S. Overall, outside of spring, MDA8 $O_3$ bias is generally lower with CMAQ531 (particularly in the eastern U.S.), with the noted exception
of increased negative bias in California.

Figure 3 presents a time series of monthly average $PM_{2.5}$ bias for all AQS sites within the 12-km domain for the CMAQ521 and five CMAQ531 simulations. Similar time series plots for $PM_{2.5}$ concentration, RMSE, and COR are in Figs. S6 – S8, along with spatial plots of seasonal $PM_{2.5}$ bias for the CMAQ521 simulation (Fig. S9). Comparing first the CMAQ521 simulation and the CMAQ531_WRF38_M3Dry_noBiDi_RWC simulation, $PM_{2.5}$ is
underestimated throughout the year with CMAQ521, with monthly average biases ranging from approximately zero to   1.5   µg   m$^{-3}$   and   the   largest   underestimations   in   the   summer.   Concentrations   of   $PM_{2.5}$   in   the CMAQ531_WRF38_M3Dry_noBiDi simulation are significantly higher (0.5 - 1.2 µg m$^{-3}$ monthly average) than the CMAQ521 simulation in winter (Dec, Jan, Feb) and slightly higher in the late summer through early fall (Aug, Sep, Oct). The higher $PM_{2.5}$ concentrations with CMAQ531  are primarily due to more abundant oxidants (e.g., $O_3$, OH$^-$)
in the winter in CMAQ531, which results in relatively large increases in OC (0.5 – 2.0 µg m$^{-3}$ monthly average) and non-carbon organic matter (NCOM) (0.5 – 1.0 µg m$^{-3}$ monthly average), primarily over the eastern half of the U.S. (not shown). When PCSOA application to RWC sources is removed (CMAQ531_WRF38_M3Dry_noBiDi), $PM_{2.5}$





concentrations are significantly reduced in the winter compared to the simulation with PCSOA applied to RWC sources, and are slightly higher than but much closer to, the monthly average $PM_{2.5}$ concentrations from the CMAQ521 simulation. The largest decrease in $PM_{2.5}$ occurs in the winter and spring, when monthly average $PM_{2.5}$ concentrations decrease by approximately 0.5 - 1.2 $\mu g \, m^{-3}$ (essentially counterbalancing the increase in $PM_{2.5}$ due to the increased oxidants in winter). Changes in monthly average $PM_{2.5}$ concentrations in summer and early fall are generally less than 0.25 $\mu g \, m^{-3}$, indicating little impact from removing the application of PCSOA on RWC sources outside of winter and spring.

Enabling BiDi (CMAQ531_WRF38_M3Dry_BiDi) increases $PM_{2.5}$ concentrations by a small amount throughout the year (Figs. 3 and S6), which is expected given that the effect of BiDi is generally localized to areas with high amounts of $NH_3$ (e.g. crop lands), so the highly spatially and temporally averaged time series plots cannot highlight the heterogenous impact of BiDi. Areas with large agricultural production requiring frequent fertilization, such as the San Joaquin Valley (SJV) in California and the mid-western U.S., are expected to show a much larger impact from implementation of the BiDi option (Pleim et al., 2019).

Figure 4 shows the difference in seasonal average $PM_{2.5}$ absolute bias between the CMAQ521 simulation and the CMAQ531_WRF38_M3Dry_BiDi simulation for AQS and NAPS sites. For winter, the change in bias is relatively small for most of the sites located in the U.S., except in the southwestern U.S. and California, where the bias is notably higher with CMAQ531, and in sites across the northern portion of the U.S. where the change in bias (both positive and negative) tends to be higher than sites in other parts of the U.S. Conversely, the NAPS sites in Canada indicate a large widespread decrease in $PM_{2.5}$ bias with CMAQ531, which may result from differences in the Canadian emission inventory (e.g. RWC) leading to a larger impact on $PM_{2.5}$ from the updates in CMAQ531. In spring, the largest differences in bias occur for sites in the upper Midwest and into the northeastern U.S., particularly for sites in and around the Great Lakes region where both large increases and decreases in bias occur, and through the Midwest states stretching down to Texas where the bias is generally higher with CMAQ531.

In summer, the most notable difference between the two simulations is a widespread, large decrease in bias in the southeastern U.S. in the CMAQ531_WRF38_M3Dry_BiDi simulation, largely the result of increased $PM_{2.5}$ from monoterpene SOA from the new AERO7 module. In late summer and early fall, total $PM_{2.5}$ increases primarily in areas rich in vegetation (e.g., southeastern U.S.) due to increased monoterpene oxidation products with secondary effects due to water uptake to particles in AERO7. In the southeastern U.S., the increase in $PM_{2.5}$ in summertime (0.75 – 3.0 $\mu g \, m^{-3}$ monthly average) is primarily driven by an increase in OC (with smaller increases during other times of year due to lower monoterpene emissions and lower oxidants in other seasons). Including additional water uptake to hydrophilic organic species results in small decreases in total $PM_{2.5}$ due to changes in the size distribution which increases dry deposition. Bias is also broadly lower in the simulation using CMAQv531 across the upper Midwest, Northwest U.S. and into southwestern Canada, while bias increases slightly across the Rocky Mountain region and parts of the upper Midwest U.S. and southern Canada. Similar to summer, there is a relatively large decrease in $PM_{2.5}$ bias in fall across the southwestern U.S., juxtaposed with several sites where bias increases. There are also relatively



large changes in bias through the upper Midwest U.S., and broadly lower bias, albeit only slightly, along the west coast states, with higher bias in the SJV of California.

Figure 5 presents seasonal stacked bar plots of observed speciated $PM_{2.5}$ for all sites in AQS that reported speciated data (includes both CSN and IMPROVE sites) and the corresponding simulated values from the various CMAQ simulations. The OTHR species (PM other) is calculated as a difference between the total measured/predicted $PM_{2.5}$ and the sum of the individual measured/predicted $PM_{2.5}$ species. There is generally good performance for $SO_4^{2-}$, $NO_3^-$ and $NH_4^+$ bias throughout the year, with all three species nearly unbiased in winter and fall, and slightly (0.25 – 0.5

$\mu g\ m^{-3}$) underestimated in spring and summer. An overestimation in OC with CMAQ521 present in winter, spring, and fall is reduced in the CMAQ531 simulations, although OC is still overestimated compared to the observations. Total $PM_{2.5}$ is grossly underestimated in summer, driven primarily by OTHR, which if removed from both the observed and modeled values would result is a much smaller $PM_{2.5}$ underestimation. More work is needed to better classify the observed "PM other" mass so that the model estimates can likewise be improved.

Overall, while the monthly average $PM_{2.5}$ concentrations are similar between CMAQ521 and CMAQ531, there are some relatively large regional and seasonal differences between the two versions of the model. In addition, while the averaged $PM_{2.5}$ concentrations are overall comparable between the two versions of the model, the underlying science has been improved in CMAQ531. Note that all the simulations presented here use BELD5 data, which is currently not available publicly. The latest BELD data available publicly for CMAQ is version 4. Using BELD5 reduces $PM_{2.5}$

concentrations, particularly in the spring and summer, relative to BELD4 due primarily to a reduction in the leaf biomass from 750 $gm^{-2}$ in BELD4 to 400 $gm^{-2}$ in BELD5 for all pine species, which results in lower SOA mass. Similarly configured CMAQ simulations using BELD4 data would likely estimate higher $PM_{2.5}$ concentrations than presented here, particularly in the spring and summer.

### 4.2 HCMAQ simulations and impact on BCs

The analysis presented in Sect. 4.1 focused on isolating the impacts of the science updates in CMAQ53 on model performance, including updating the science in the lateral BCs by transitioning from BCs created from a HCMAQ521 simulation to those created from a HCMAQ53 simulation. Lateral BCs increasingly influence pollutant concentrations in the regional simulation, as locally emitted pollutants across the U.S. have declined over time. Pollutants specified in the mid and upper troposphere in the BCs can be advected long distances and interact with locally emitted pollutants

to alter surface and boundary layer pollutant concentrations (Hogrefe et al., 2018). As described in Sect. 3.3, BCs for the regional CMAQ simulations were derived from a HCMAQ521 simulation (used for the regional CMAQ521 simulation) and from a HCMAQ53 simulation using the M3Dry deposition model (used for the CMAQ531 M3Dry simulations). In this section we briefly compare the operational performance of the two HCMAQ simulations and examine how differences in HCMAQ model performance influence the BCs used in the regional-scale CMAQ

simulations.

Figure 6 shows the difference in seasonal average surface $O_3$ (all hours) between the HCMAQ521 and HCMAQ53_M3Dry simulations. Analogous to the regional CMAQ simulations, $O_3$ mixing ratios are consistently



higher (10 – 20 ppbv) in the HCMAQ53_M3Dry simulation in the winter and spring over the higher latitudes of North America where snow cover is more prevalent. In the mid and lower latitudes, $O_3$ mixing ratios are generally lower in

the HCMAQ53_M3Dry simulation by approximately 0.0 – 5.0 ppbv in winter to 5.0 – 10 ppbv in fall, due in large part to the updates to bromine/iodine chemistry that result in lower $O_3$ mixing ratios over the oceans. These differences consistently reduce $O_3$ in the mid to lower latitudes with CMAQ53 derived lateral BCs compared to CMAQ521 derived BCs, which contributes to reducing MDA8 $O_3$ in spring, summer, and fall in the regional CMAQ531 simulations. Although not shown in Fig. 6, the lower $O_3$ in the HCMAQ53_M3Dry simulation extends through the

lower troposphere (Sarwar et al., 2019).

Figure 7 shows the difference in seasonal average surface $PM_{2.5}$ (all hours) between the HCMAQ521 and HCMAQ53_M3Dry simulations. The pattern mimics that in the regional CMAQ simulations, with a relatively small difference over and around North America in winter and larger differences in spring, summer, and fall. In spring, the differences start to reveal higher $PM_{2.5}$ in the southeastern U.S. and slightly lower $PM_{2.5}$ in the west. In summer, there

is a large increase in $PM_{2.5}$ in the southeast and north along the east coast of the U.S. and a small increase in the west with CMAQ53. In fall, the increase in $PM_{2.5}$ in the HCMAQ53_M3Dry simulation is still present in the southeastern U.S. Outside the conterminous U.S., the difference in $PM_{2.5}$ is relatively small between the two HCMAQ simulations, suggesting that differences in simulated $PM_{2.5}$ concentrations in the regional CMAQ simulations for the U.S. and Canada are minimally influenced by differences between the BCs.

**4.3 Impact of updating meteorology**

Meteorology is a critical component of air quality modeling, as changes in meteorology have the potential to drastically alter pollutant concentrations over large temporal and spatial scales, consequently impacting overall performance of the air quality modeling system. Meteorology models are frequently updated to improve underlying science, add functionality, and address errors in the model. As described in Sect. 3.1, two versions of WRF (WRF38

and WRF411) were used during the development and testing of the CMAQ53 modeling system. In this section we compare the operational performance between two CMAQ531 simulations with the same configuration except that one is driven using WRF38 (SVC) inputs and the other is driven using WRF411 (HVC). Although the HCMAQ simulation from which BCs are derived was not run using WRF411, they have been reprocessed to the WRF HVC to align with the vertical coordinate of the CMAQ simulation that was run using WRF411. This reprocessing results in

only a very small difference between BCs using the SVC and HVC. Note that versions of CMAQ prior to 5.3 and versions of MCIP prior to 5.0 are not compatible with the WRF HVC.

MDA8 $O_3$ is consistently higher throughout the year in the simulation using WRF411 (CMAQ531_WRF411_M3Dry_BiDi), with the largest increase occuring in spring and fall when monthly averaged MDA8 $O_3$ is approximately 1.5 ppbv higher on average (Fig. 1). The increase is slightly smaller (1.0 – 1.5 ppbv) in

winter and summer. This increase in $O_3$ generally reduces bias in the simulation using WRF411 throughout most of the year, except during warmer months (Jul, Aug, Sep) when bias increases, as MDA8 $O_3$ is overestimated on average in summer. Figure 8 shows seasonal average spatial plots of the change in MDA8 $O_3$ absolute bias at all AQS and NAPS sites. Sites in the southern U.S. (particularly along coastal areas), California, and the Northwest show the largest





reduction in bias in winter, while sites in the upper Midwest show the largest increase in bias. In spring, almost all the sites indicate some reduction in bias with WRF411, except along the southern tip of Texas and scattered sites in the Northwest. In summer, the pattern shifts, with sites in the western U.S. and Great Lakes regions showing the largest decrease in bias, while sites in the upper Midwest still indicate an increase in bias with WRF411. Fall has the largest mixed signal in bias change, with sites in the southern U.S. and California indicating a decrease in bias, while sites in the upper Midwest and Northeast indicate a relatively large increase.

As explained in Sect. 3.1, the WRF411 simulations used vegetation parameters directly from MODIS satellite retrievals which result in generally lower values for VF and LAI, especially in most of western North America (Fig. S1). These differences not only affect the meteorological simulations but also the chemical surface fluxes (dry deposition and bidirectional fluxes) since these algorithms in CMAQ use the vegetation parameters from WRF (via the MCIP input files). The higher MDA8 $O_3$ concentrations evident in the time series (Fig. 1) and spatial plot (Fig. 8) for the simulation using WRF411 are primarily due to reduced dry deposition of $O_3$ as deposition to vegetation is generally greater than deposition to bare ground (Fig. S10). Thus, the lower VF and LAI from the WRF411 simulation result in lower $O_3$ deposition velocities and therefore higher ambient concentrations. Also potentially contributing to the higher $O_3$ mixing ratios with WRF411 are systematically drier (lower precipitation) conditions with WRF411 versus WRF38 throughout the year, particularly in the central and eastern U.S., where WRF tends to underestimate the observed precipitation (Figs. S12 – S15). The lower precipitation in WRF411 results in less $O_3$ wet scavenging and may also reflect fewer fair-weather cumulus clouds, which would increase $O_3$ photochemical production.

For $PM_{2.5}$, transitioning from WRF38 to WRF411 is largely unremarkable. Apart from January and February, $PM_{2.5}$ concentrations are consistently higher throughout the year in the simulation using WRF411, particularly in summer and fall when monthly average values increase by approximately $0.1 – 0.25$ µg m$^{-3}$ (Fig. 3). Figure 9 shows seasonal average spatial plots of the change in $PM_{2.5}$ absolute bias between the two simulations. In winter, there are widespread changes in $PM_{2.5}$ bias, while in spring sites in the southeastern U.S. and Mid-Atlantic show larger decreases in bias with WRF411 compared to sites across the rest of the country. In summer, sites in the eastern half of the U.S. indicate the largest change in bias, while sites in the western half of the country generally show a small decrease in bias with WRF411. Finally, for fall, there is generally a decrease in bias with WRF411, particularly at sites in the southeastern U.S., lower Midwest, and Rocky Mountain regions. The differences in $PM_{2.5}$ concentrations using both versions of WRF are likely attributed to a combination of the lower precipitation in WRF411 and the differences in vegetation parameters (Fig. S11). Drier conditions reduce aerosol growth, which reduces deposition velocities and settling velocities for the larger size particles. The changes in vegetation parameters affect $PM_{2.5}$ because, as noted above, the aerosol dry deposition model now has dependencies on LAI and VF.

**4.4 Impact of STAGE and M3Dry deposition models**

The final updates in CMAQ531 examined are the updated M3Dry and new STAGE deposition models. While both deposition models originate from previous versions of the M3Dry model which has a long history in CMAQ and CTMs (Pleim et al., 1984), the dry deposition algorithms in the updated M3Dry and STAGE models differ substantially in many of their resistance algorithms, which substantially alters simulated pollutant concentrations and



depositions. Here, CMAQ531 simulations are compared using the same configuration except for the deposition model (i.e. M3Dry or STAGE). To consistently represent air-surface exchange processes across scales, the lateral BCs for the regional M3Dry and STAGE simulations are generated from HCMAQ simulations configured with the corresponding deposition model.

       The time series of monthly average MDA8 $O_3$ bias across all AQS sites for the M3Dry
(CMAQ531_WRF411_M3Dry_BiDi) and STAGE (CMAQ531_WRF411_STAGE_BiDi) simulations are presented in Fig. 1. Aside from several warmer months (Jul, Aug, Sep), when both simulations have very similar performance, MDA8 $O_3$ is consistently higher in the M3Dry simulation. In spring, monthly average MDA8 $O_3$ mixing ratios are 1.0 – 1.5 ppbv higher in the M3Dry simulation, while in Nov and Dec mixing ratios are 1.5 – 2.0 ppbv higher. The higher mixing ratios in the M3Dry simulation improve overall performance for MDA8 $O_3$ bias versus the STAGE simulation,
as MDA8 $O_3$ is broadly underestimated by CMAQ outside of summer. Figure 10 shows the difference in seasonal MDA8 $O_3$ absolute bias between the M3Dry and STAGE simulations. For winter, the simulation using M3Dry shows improved (smaller) bias across most of the sites, except for the upper Midwest and Great Lakes regions where the simulation using STAGE has smaller bias. In spring, almost all sites indicate smaller bias with M3Dry, while in summer the bias signal is generally mixed with no apparent pattern. In fall, the simulation using STAGE has smaller
bias for sites across the northern portion of the U.S. and Canada (note that sites in Canada are not included in the time series plots), while the simulation using M3Dry has smaller bias for sites across the southern U.S and California. The generally higher $O_3$ in the M3Dry simulation is likely a combination of both higher $O_3$ in the regional simulation and higher $O_3$ introduced from the lateral BCs.

       For daily average $PM_{2.5}$, the simulation using STAGE has consistently higher monthly average concentrations
(approximately 0.25 – 0.75 µg m$^{-3}$) throughout the year, which, except during Apr, Sep, and Oct, reduces bias (Fig. 3).  For winter, sites in the eastern U.S. generally have smaller bias with M3Dry while sites in the western U.S. tend to have smaller bias with STAGE (Fig. 11). The pattern shifts slightly in spring and summer, with sites in the Great Lakes, Northeast and western U.S. having smaller bias with M3Dry while sites in the southern U.S. and Canada have smaller bias with STAGE. In fall, sites in the northern U.S. and Canada have smaller bias with STAGE while sites in
the southern and western U.S. have smaller bias with M3Dry. Overall, the difference in bias for $PM_{2.5}$ between the two deposition models is consistent throughout the year with relatively similar patterns in bias regardless of season.

       The consistently higher $PM_{2.5}$ concentrations using the STAGE model are due to differences in the assumptions affecting deposition velocity to surfaces in the M3Dry and STAGE models. M3Dry uses a revised formulation of the dry deposition impaction term so that it integrates the effect of mode width in the Stokes number via the settling
velocity rather than adjusting the impaction term magnitude directly, which intends to resolve massive overprediction of deposition velocity for coarse-mode particles. Also, the impaction term in M3Dry in CMAQ53 is based on Slinn (1982), while the Stokes number is based on Slinn (1982) and Giorgi (1986). In STAGE, the Stokes number for each vegetated land-use type is calculated using the leaf geometry from Zhang et al. (2001), and a different form of the Stokes number is used for deposition to smooth surfaces, ground, and water. These different parametrizations
implemented in M3Dry and STAGE lead to the different estimates for $PM_{2.5}$ (and $O_3$).





For ambient $NH_3$, M3Dry and STAGE perform similarly throughout the year, with some notable exceptions. The time series in Fig. 12 shows the observed monthly average $NH_3$ concentration from the AMON network sites along with the corresponding simulated $NH_3$ concentrations and monthly mean bias, RMSE, and Pearson correlation coefficient for the M3Dry and STAGE simulations. While both bidirectional models capture the overall trend in observed $NH_3$,

with lower concentrations in winter that peak in summer, both models underestimate the observed monthly average $NH_3$ concentrations by $0.2 – 1.0 \mu g \ m^{-3}$. The STAGE model performs slightly better outside of summer, with lower bias and RMSE from January through May and October through November and higher correlation throughout the year. In summer, M3Dry has smaller bias, roughly half that of STAGE for July through September, but higher RMSE and lower correlation. The seasonal average $NH_3$ spatial plots (Fig. 13) show little difference in absolute bias between

M3Dry and STAGE in winter and relatively small differences in spring, except at two sites in the west that have significantly smaller ($>1.0 \mu g \ m^{-3}$) bias with STAGE and one site in the central SJV of California that has smaller bias ($0.5 \mu g \ m^{-3}$) with M3Dry. The difference in bias for the remaining sites is generally less than $0.1 \mu g \ m^{-3}$. In summer and fall, $NH_3$ bias is lower ($0.1 – 0.3 \mu g \ m^{-3}$) across most sites in the Northeast and Mid-Atlantic with M3Dry, while most sites west of the Mississippi River show a relatively small ($< 0.1 \mu g \ m^{-3}$) difference in bias between the two

models. There are some notable exceptions, such as several sites in southern Virginia and North Carolina where STAGE has significantly smaller ($0.5 – 2.0 \mu g \ m^{-3}$) bias, as well as sites in Arkansas, Texas, and Wyoming, while M3Dry maintains smaller bias for sites in the SJV of California and several sites in the Midwest.

### 4.5 CMAQv5.3.1 operational model performance summary

A brief summary of the operational performance (Dennis et al., 2010) is presented for the

CMAQ531_WRF411_M3Dry_BiDi simulation by examining several species. The CMAQ531_WRF411_M3Dry_BiDi simulation is shown since it utilizes the most updated versions of both WRF and CMAQ used here. Figures 10 and 11 present range values of MB, NMB, RMSE, and COR for MDA8 $O_3$ and $PM_{2.5}$, respectively, for all AQS sites for the CMAQ531_WRF411_M3Dry_BiDi simulation computed for each season and NOAA climate region (Fig. S16; https://www.ncdc.noaa.gov/monitoring-references/maps/us-climate-regions.php).

Similar figures for OC, $NO_3^-$, and $SO_4^{2-}$ for multiple networks are presented in Figs. S17 – S27.

Figure 14 highlights the widespread underestimation of $O_3$ with CMAQ531 in spring for all regions, while outside of spring MDA8 $O_3$ is consistently underestimated in the Southwest and West regions and overestimated in the Ohio Valley and Upper Midwest regions. Underestimated $O_3$ from the lateral BCs likely contributes strongly to the large springtime, and smaller wintertime, underestimation of MDA8 $O_3$. Sarwar et al. (2019) showed that for a 2006

HCMAQ simulation CMAQ underestimated $O_3$ mixing ratios at CASTNet sites (which better represent long-range transported (LRT) $O_3$) in spring and early summer, while a similar analysis for the HCMAQ53_M3Dry and HCMAQ53_STAGE simulations show a relatively large ($5.0 – 8.0$ ppbv monthly average) underestimation $O_3$ for CASTNet sites for March through June. Since springtime $O_3$ at the majority of the rural CASTNET monitor locations is likely driven by LRT $O_3$, this underestimation is likely associated with the underestimation in the large-scale $O_3$

distributions in the lower troposphere of the HCMAQ simulations used in this analyses, which subsequently is influenced by uncertainties in global emission estimates, representation of $O_3$ depletion in LRT air masses as they



traverse large oceanic regions, and representation of stratosphere-troposphere exchange processes. This hypothesis is supported by comparisons of tropospheric $O_3$ mixing ratios from the HCMAQ53_M3Dry to ozonesonde observations from 20 World Ozone and Ultraviolet Radiation Data Centre (WOUDC) sites which show that $O_3$ is consistently underestimated by 20 – 40 ppbv in the mid- to upper troposphere (250 – 100 hPa) at most sites in spring (Figs. S28 – S32).

Figure 15 highlights a general underestimation of $PM_{2.5}$ in the western U.S. for the CMAQ531_WRF411_M3Dry_BiDi simulation. These underestimates are largely attributed to deficiencies in the meteorology simulated by WRF, which did not reproduce the strong wintertime inversions that occur in the western U.S.; higher-resolution simulations may better resolve cold pools in the complex terrain in those regions (Kelly et al., 2019). $PM_{2.5}$ in the eastern U.S. is mostly unbiased throughout the year, with higher correlation and lower RMSE than the western U.S. regions. The categorical performance for OC (Figs. S17 – S19) is relatively good for most regions and seasons, with some exceptions. Organic carbon is overestimated in the Northeast, Ohio Valley, and Upper Midwest regions in spring, with smaller overestimations in the other seasons, and underestimated in the Northern Rockies and Plains region in spring and summer. The underestimation of OC (SOA) in summer in the southeastern U.S. noted previously (Appel et al., 2017; Murphy et al., 2017; Xu et al., 2018; Zhang et al., 2018) has been eliminated in CMAQ531 and replaced with a small overestimation. Particulate $NO_3^-$ is underestimated for most seasons and regions (Figs. S20-S23), with the largest underestimation in the western U.S. in spring and summer. Performance for particulate $SO_4^{2-}$ is generally good for most regions and seasons (Fig. S24), except in the Northwest region where NMB values exceed 60% throughout the year (although MB values are < 0.4 ug m$^{-3}$).

## 5 Discussion

CMAQ53 was publicly released in August 2019 and followed shortly by CMAQ531 in December 2019. The major science updates in the new model have been described, including extensive chemistry updates; the new AERO7 aerosol module; the updated M3Dry bidirectional deposition model; and the new STAGE bidirectional deposition model. Other significant updates to the CMAQ system include support for the HVC in WRF, updates to the PX-LSM and ACM2 PBL schemes in WRF and CMAQ, and the new DESID system. Evaluation of the science updates in CMAQ53 was accomplished by comparing monthly and seasonal MDA8 $O_3$ and $PM_{2.5}$ values from CMAQ531 simulations to a similarly configured CMAQ521 simulation for 2016. For MDA8 $O_3$, CMAQ531 has consistently higher $O_3$ in the winter versus CMAQ521, primarily due to reduced dry deposition to snow, and lower $O_3$ throughout the rest of the year, particularly in spring due in large part to reduced $O_3$ in both the regional simulation and originating from the lateral BCs. The result is generally reduced MDA8 $O_3$ bias in winter and summer and increased bias in spring and fall with CMAQ531. For $PM_{2.5}$, CMAQ531 has lower concentrations on average in the spring and fall, higher concentrations in summer, and essentially unchanged concentrations in winter compared to CMAQ521. Overall for $PM_{2.5}$, bias is slightly increased in spring and fall and slightly reduced in summer.

Comparisons were also made for the lateral BCs, meteorology model version, and the bi-directional surface exchange model to quantify their impacts on model results. Updating the source of lateral BCs from a HCMAQ521 simulation



to a HCMAQ53 simulation increases $O_3$ mixing ratios from the BCs in the northern latitudes (especially in winter and spring) due to the decreased $O_3$ dry deposition to snow in CMAQ53, and reduces $O_3$ mixing ratios in the mid and lower latitudes due to the updates to the bromine/iodine chemistry in CMAQ53 which reduces $O_3$ mixing ratios over the ocean. Transitioning from WRF38 using the SVC to WRF411 using the HVC in the 12-km domain consistently increases (1.0 – 1.5 ppbv) MDA8 $O_3$ mixing ratios throughout the year, while the impact on $PM_{2.5}$ is smaller but also with consistently higher concentrations (0.1 – 0.25 μg m$^{-3}$) throughout the year. The differences in pollutant concentrations in CMAQ from using WRF38 and WRF411 are primarily attributed to differences in the treatment of VF and LAI in the LSM in the two version of WRF. Using MODIS derived vegetation parameters in WRF411 results in overall lower values of VF and LAI compared to the NLCD derived values in WRF38, particularly in western North America. Finally, for the M3Dry and STAGE models, MDA8 $O_3$ is similar in summer between the two models but generally higher with M3Dry outside of summer. For $PM_{2.5}$, STAGE has consistently higher concentrations throughout the year, driven by differences in the assumptions affecting deposition velocity of particles in the two models. For ambient $NH_3$, STAGE has slightly higher concentrations and smaller bias in spring and fall, M3Dry has higher concentrations and smaller bias in summer, and both models have relatively similar concentrations in winter. Model error tends to be lower and correlation higher for $NH_3$ throughout the year with STAGE versus M3Dry.

While these versions of CMAQ modeling system represent significant advancement in model process and input data science, several performance issues remain. The large underestimation of $O_3$ in spring highlights the need for further improvement in representing the impacts of large-scale $O_3$ distributions (e.g., international emissions, marine chemistry, stratosphere-troposphere exchange) and dry deposition across different surfaces which influence low- to mid-level $O_3$ mixing ratios simulated by the model. Ozone continues to be underestimated in California throughout the year, a longstanding issue previously noted by Appel et al. (2017). The persistent underestimation is likely due to inaccurate representation of California emissions in the inventory used, as well as inability of the current 12-km resolution to capture atmospheric dynamics in the complex terrain in the region. CMAQ's performance in representing spatial and seasonal variations in ambient $PM_{2.5}$ has improved considerably over the last decade as a result of improvements in representation of PM formation pathways and emissions, however CMAQ still tends to underestimate total $PM_{2.5}$ mass in the southern and western U.S. throughout much of the year, driven primarily by a large underestimation of $PM_{other}$ followed by a smaller underestimation of $NO_3^-$. Although efforts have been made to quantify the species that comprise the unidentified $PM_{2.5}$ mass in the observations, more work is required to further improve the emission inventories for primary PM and consequently model performance.

**Code availability**

The CMAQ version 5.3 (https://doi.org/10.5281/zenodo.3379043) and 5.3.1 (https://doi.org/10.5281/zenodo.3585898) codes and MCIP version 4.5 and 5.0 codes are available from the CMAQ Github site (https://github.com/USEPA/CMAQ). The AMET code is available from the AMET Github site (https://github.com/USEPA/AMET).



**Data availability**

All data used in this work are available upon request from the authors. Please contact the corresponding author to request any data related to this work.

**Author contribution**

905 K. Wyat Appel lead the development of this manuscript and was responsible for most of the model evaluation components presented in Section 4. Authors Jesse O. Bash, Kathleen M. Fahey, Robert C. Gilliam, William Hutzell, Daiwen Kang, Rohit Mathur, Benjamin Murphy, Sergey Napelenok, Christopher Nolte, Jonathan E. Pleim, George Pouliot, Havala O. T. Pye, Limei Ran, Shawn J. Roselle, Golam Sarwar, Donna B. Schwede, Fahim Sidi, and David Wong contributed to the CMAQv5.3 code development and also contributed directly to the writing of Section 2 of

910 this manuscript. Authors Kristen M. Foley and Christian Hogrefe contributed to the writing of Section 3 and the model evaluation components of this study. Tanya L. Spero contributed to the development of MCIPv5.0 discussed in Section 2.

**Competing Interests**

The authors declare no competing interests.

**Disclaimer**

The views expressed in this article are those of the authors and do not necessarily represent the views or policies of the U.S. Environmental Protection Agency.

**Acknowledgments**

Thanks to Lara Reynolds with General Dynamics Information Technology, Inc (GDIT) for performing the WRF38
simulation used in this work. Special thanks to Liz Adams with the University of North Carolina (UNC) for her extensive testing of the CMAQ53 modeling system. Special thanks to Deborah Luecken, now retired from EPA, for her significant contributions to the development of CMAQ53 and her decades of tireless work on the CMAQ modeling system.

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

**Table 1. Names and configurations of the hemispheric and regional CMAQ simulations.**

| Simulation Name | CMAQ Version | WRF Version | Emissions Version | Chemical Mechanism | PCSOA/ SVPOA | Deposition Model | BiDi | Wind Blown Dust |
|---|---|---|---|---|---|---|---|---|
| HCMAQ521 | 5.2.1 | 3.8 | HTAPv2 | CB6r3m_ae6 | No | M3Dry | No | Yes |
| HCMAQ53_M3Dry | 5.3 | 3.8 | HTAPv2 | CB6r3m_ae7_kmtbr | No | M3Dry | No | Yes |
| HCMAQ53_STAGE | 5.3 | 3.8 | HTAPv2 | CB6r3m_ae7_kmtbr | No | STAGE | No | Yes |
| CMAQ521 | 5.2.1 | 3.8 | 2016 v1 | CB6r3_ae6 | Yes | M3Dry | No | No |
| CMAQ531_WRF38_M3Dry_noBiDi_RWC | 5.3.1 | 3.8 | 2016 v1 | CB6r3_ae7 | Yes | M3Dry | No | No |
| CMAQ531_WRF38_M3Dry_noBiDi | 5.3.1 | 3.8 | 2016 v1 | CB6r3_ae7 | Yes (no RWC) | M3Dry | No | No |
| CMAQ531_WRF38_M3Dry_BiDi | 5.3.1 | 3.8 | 2016 v1 | CB6r3_ae7 | Yes (no RWC) | M3Dry | Yes | No |
| CMAQ531_WRF411_M3Dry_BiDi | 5.3.1 | 4.1.1 | 2016 v1 | CB6r3_ae7 | Yes (no RWC) | M3Dry | Yes | No |
| CMAQ531_WRF411_STAGE_BiDi | 5.3.1 | 4.1.1 | 2016 v1 | CB6r3_ae7 | Yes (no RWC) | STAGE | Yes | No |

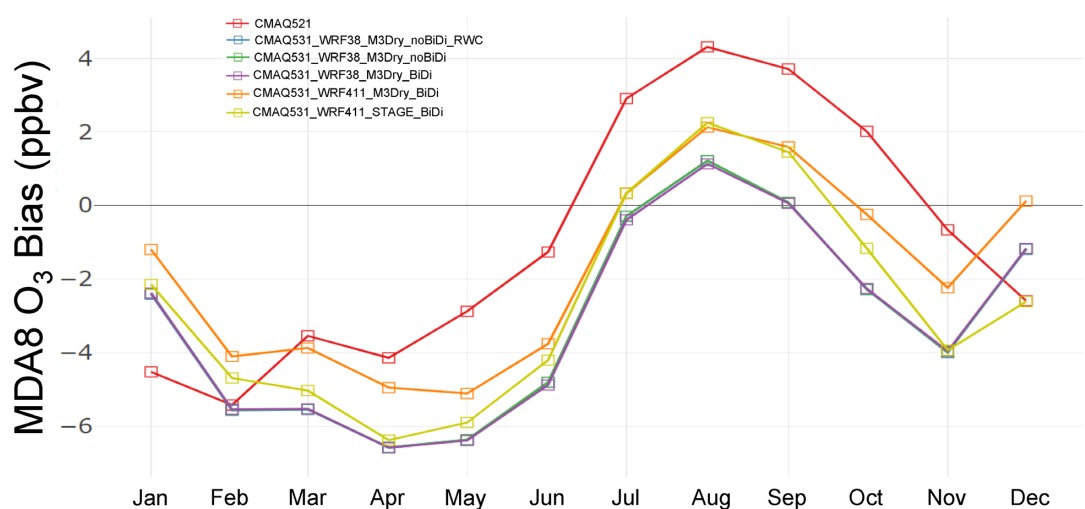

**Figure 1. Time series of monthly average MDA8 O₃ bias (ppbv) for all AQS sites for CMAQ521 (red), CMAQ531_WRF38_M3Dry_noBiDi_RWC (blue), CMAQ531_WRF38_M3Dry_noBiDi (green), CMAQ531_WRF38_M3Dry_BiDi (purple), CMAQ531_WRF411_M3Dry_BiDi (orange), and CMAQ53_WRF411_STAGE_BiDi (yellow).**






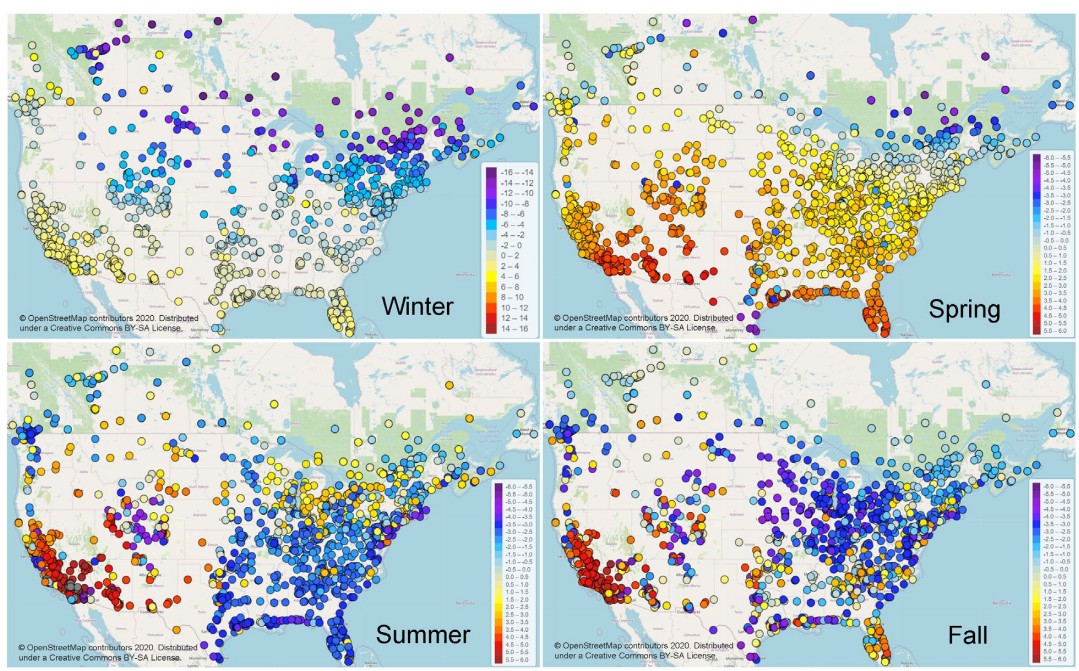

**Figure 2. Seasonal average MDA8 O₃ absolute bias difference (ppbv) between the CMAQ521 and CMAQ531_WRF38_M3Dry_BiDi simulations for all AQS and NAPS sites. Cool shading (negative values) indicate smaller bias in the CMAQ531_WRF38_M3Dry_BiDi simulation, warm colors (positive values) indicate smaller bias in the CMAQ521 simulation. Note that the scale is different for the winter season. Grey shading indicates values beyond the range of the scale.**

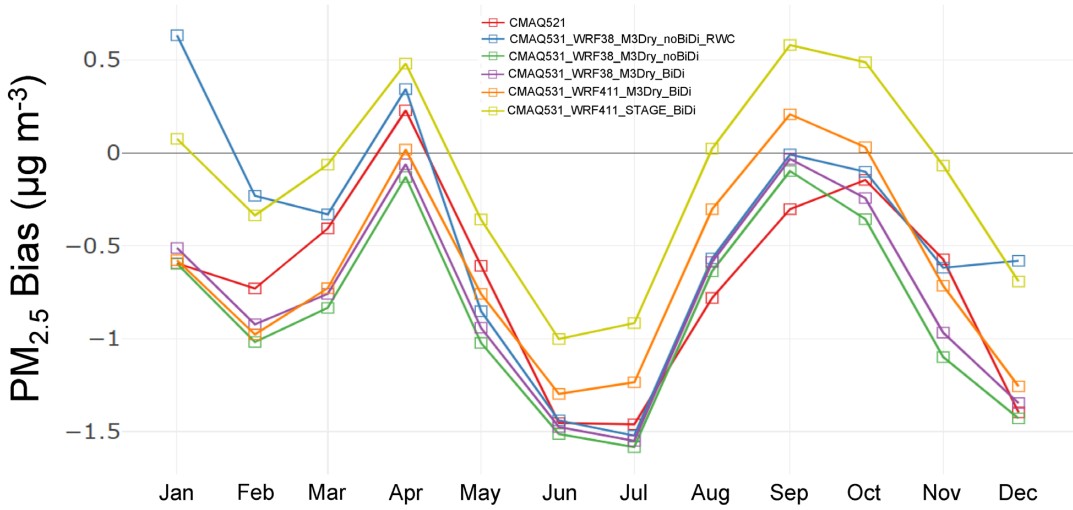

**Figure 3. Time series of monthly average PM₂.₅ bias (µg m⁻³) for all AQS sites for CMAQ521 (red), CMAQ531_WRF38_M3Dry_noBiDi_RWC (blue), CMAQ531_WRF38_M3Dry_noBiDi (green), CMAQ531_WRF38_M3Dry_BiDi (purple), CMAQ531_WRF411_M3Dry_BiDi (orange), and CMAQ531_WRF411_STAGE_BiDi (yellow).**



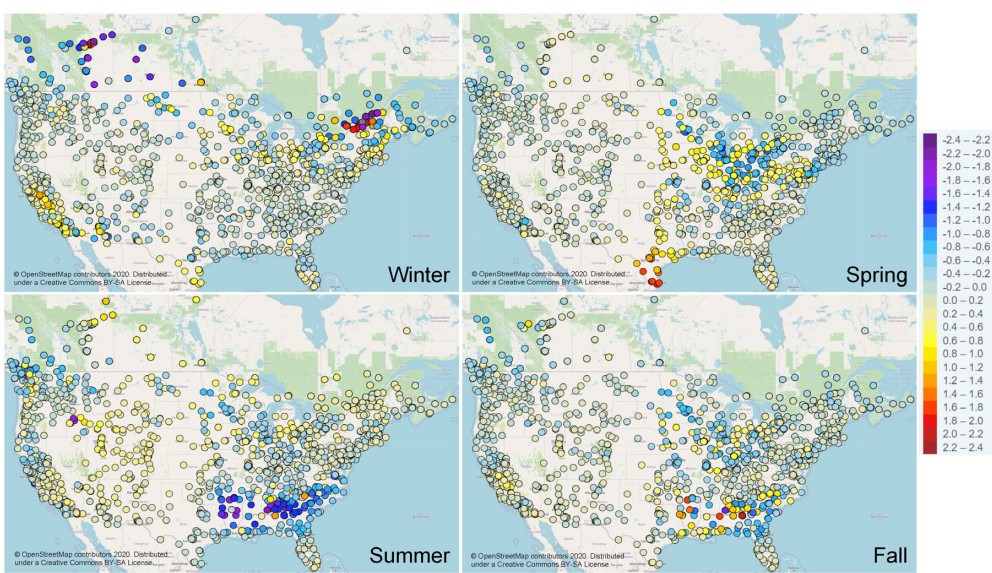

**Figure 4. Seasonal average PM$_{2.5}$ absolute bias difference (μg m$^{-3}$) between the CMAQ521 and CMAQ531_WRF38_M3Dry_BiDi simulations for all AQS and NAPS sites. Cool shading (negative values) indicate smaller bias in the CMAQ531_WRF38_M3Dry_BiDi simulation, warm colors (positive values) indicate smaller bias in the CMAQ521 simulation. Grey shading indicates values beyond the range of the scale.**

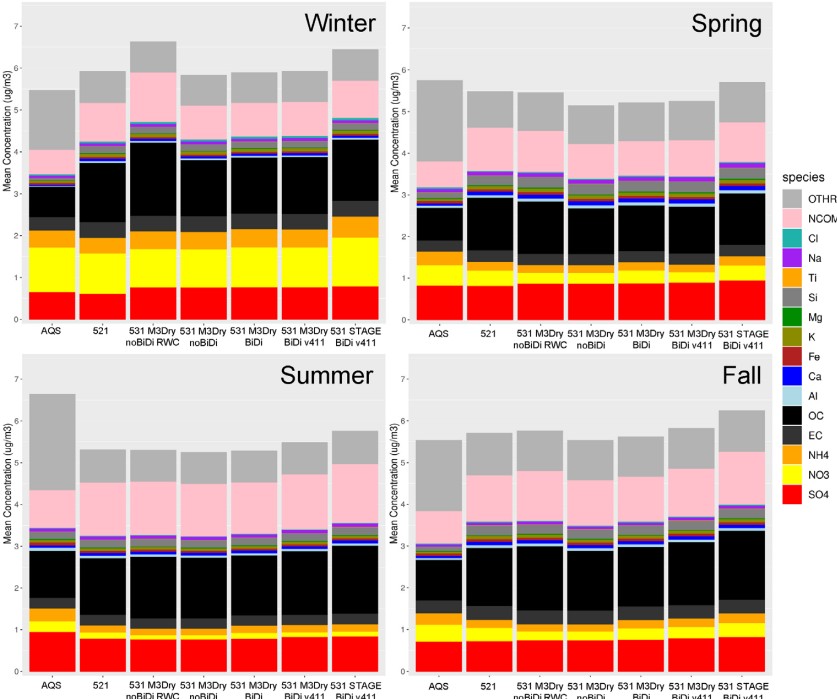

**Figure 5. Seasonal stacked bar plots of speciated PM$_{2.5}$ (μg m$^{-3}$) for AQS sites. The height of the bar represents the total PM$_{2.5}$ concentration. Refer to Table 1 for more details on the simulation configurations.**



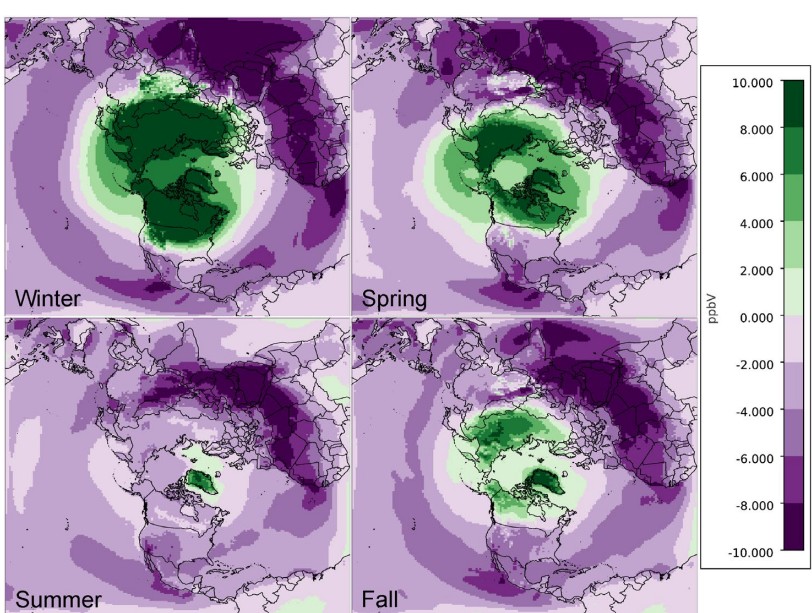

**Figure 6. Difference in seasonal average O₃ mixing ratios (ppbv; all hours) between HCMAQ521 and HCMAQ531_M3Dry simulations (HCMAQ531_M3Dry – HCMAQ521). Green shading indicates higher O₃ mixing ratios and purple shading indicates lower O₃ mixing ratios in the HCMAQ531_M3Dry simulation.**


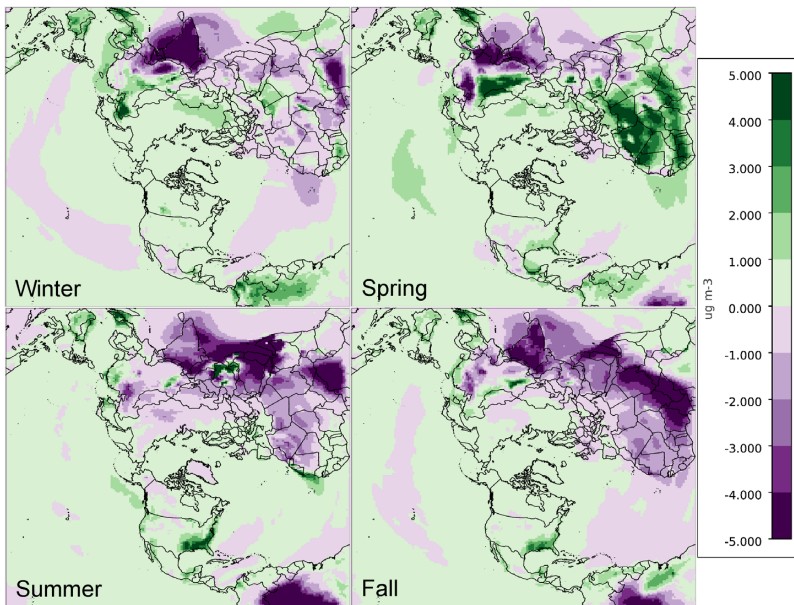

**Figure 7. Difference in seasonal average PM₂.₅ concentrations (μg m⁻³; all hours) between the HCMAQ521 and HCMAQ531_M3Dry simulations (HCMAQ531_M3Dry – HCMAQ521). Green shading indicates higher PM₂.₅ concentrations and purple shading indicates lower PM₂.₅ concentrations with the HCMAQ531_M3Dry simulation.**



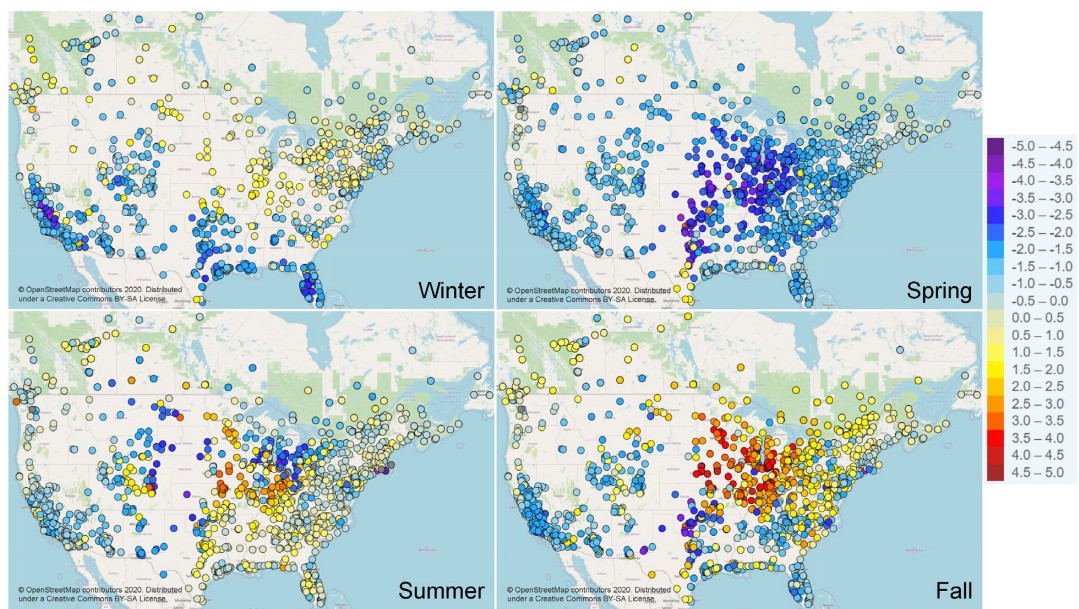


**Figure 8. Seasonal average MDA8 O₃ absolute bias difference (ppbv) between the CMAQ531_WRF38_M3Dry_BiDi and CMAQ531_WRF411_M3Dry_BiDi simulations for all AQS and NAPS sites. Cool shading (negative values) indicate smaller bias in the CMAQ531_WRF411_M3Dry_BiDi simulation, warm colors (positive values) indicate smaller bias in the CMAQ531_WRF38_M3Dry_BiDi simulation.**

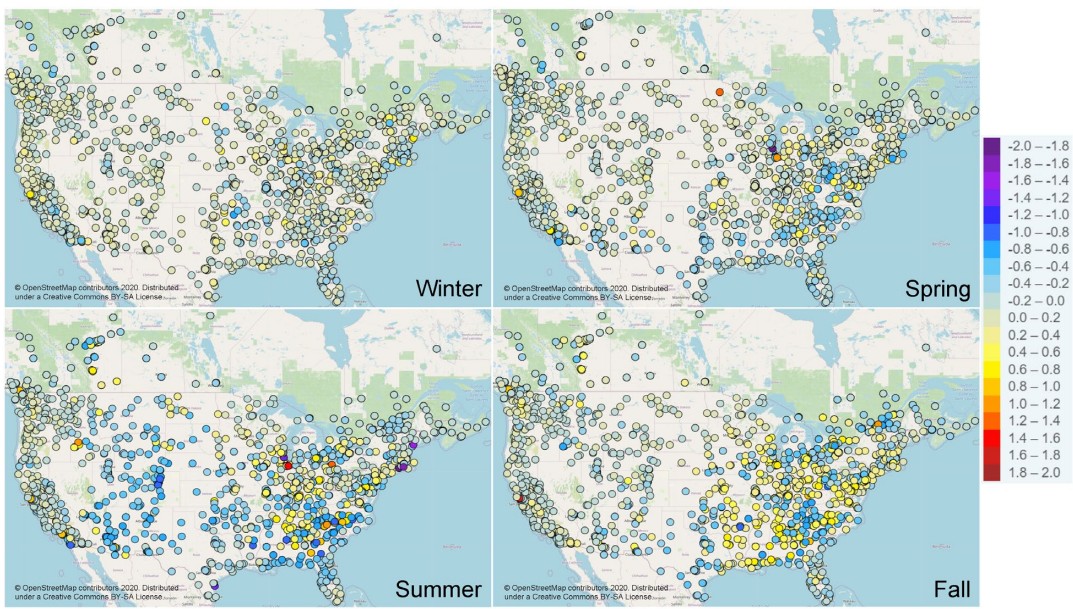


**Figure 9. Seasonal average PM₂.₅ absolute bias difference (μg m⁻³) between the CMAQ531_WRF38_M3Dry_BiDi and CMAQ531_WRF411_M3Dry_BiDi simulations for all AQS and NAPS sites. Cool shading (negative values) indicate smaller bias in the CMAQ531_WRF411_M3Dry_BiDi simulation, warm colors (positive values) indicate smaller bias in the CMAQ531_WRF38_M3Dry_BiDi simulation.**



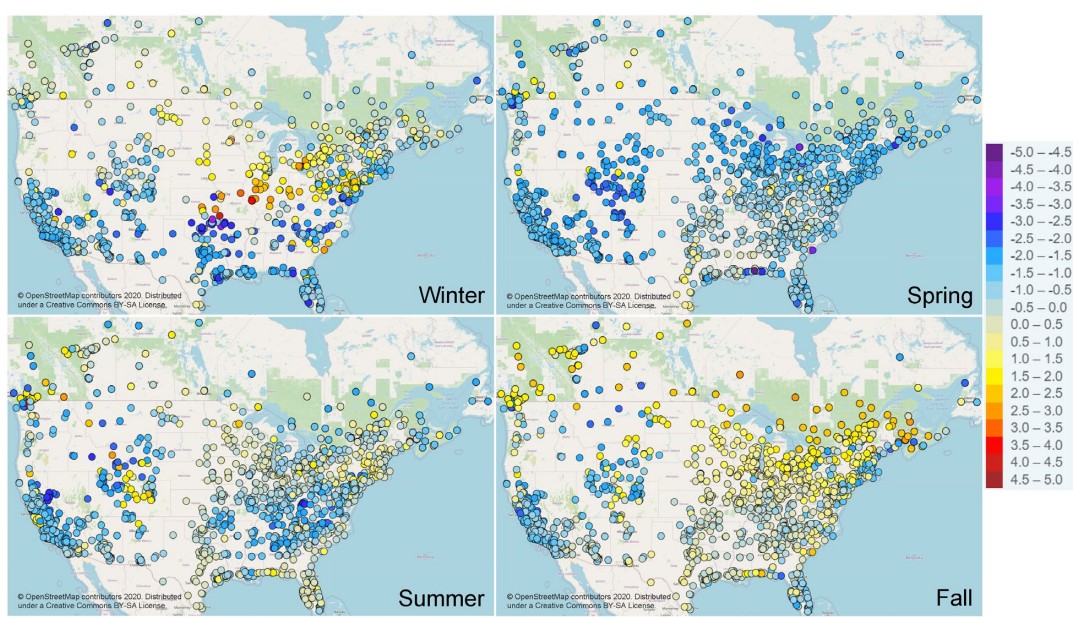


**Figure 10. Seasonal average MDA8 O₃ absolute bias difference (ppbv) between CMAQ531_WRF411_M3Dry_BiDi and CMAQ531_WRF411_STAGE_BiDi simulations for all AQS and NAPS sites. Cool shading (negative values) indicate smaller bias in the CMAQ531_WRF411_M3Dry_BiDi simulation, warm colors (positive values) indicate smaller bias in the CMAQ531_WRF411_STAGE_BiDi simulation.**

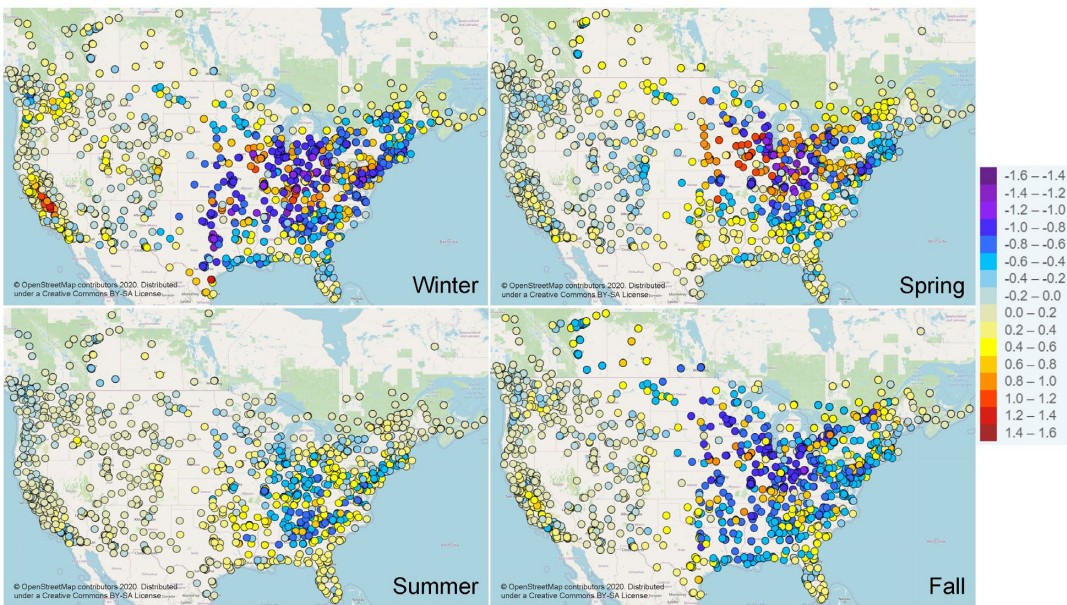


**Figure 11. Seasonal average PM₂.₅ absolute bias difference (μg m⁻³) between CMAQ531_WRF411_M3Dry_BiDi and CMAQ531_WRF411_STAGE_BiDi simulations for all AQS and NAPS sites. Cool shading (negative values) indicate smaller bias in the CMAQ531_WRF411_M3Dry_BiDi simulation, warm colors (positive values) indicate smaller bias in the CMAQ531_WRF411_STAGE_BiDi simulation.**



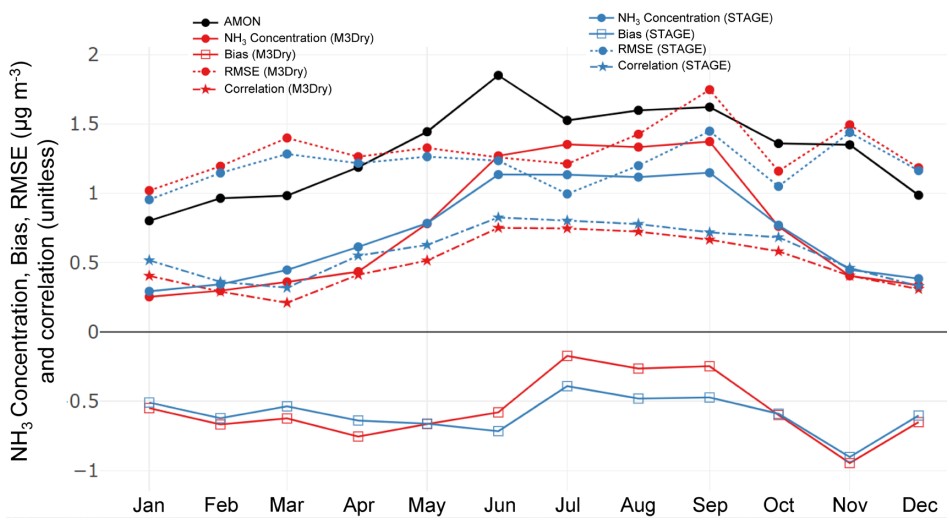


**Figure 12. Time series of monthly average observed and simulated NH₃ concentrations (solid circle; µg m⁻³), bias (open square; µg m⁻³), RMSE (dotted solid circle; µg m⁻³), and Pearson correlation coefficient (dot-dash solid stars) for all AMON sites for CMAQ531_WRF411_M3Dry_BiDi simulation (red) and CMAQ531_WRF411_STAGE_BiDi simulation (blue).**

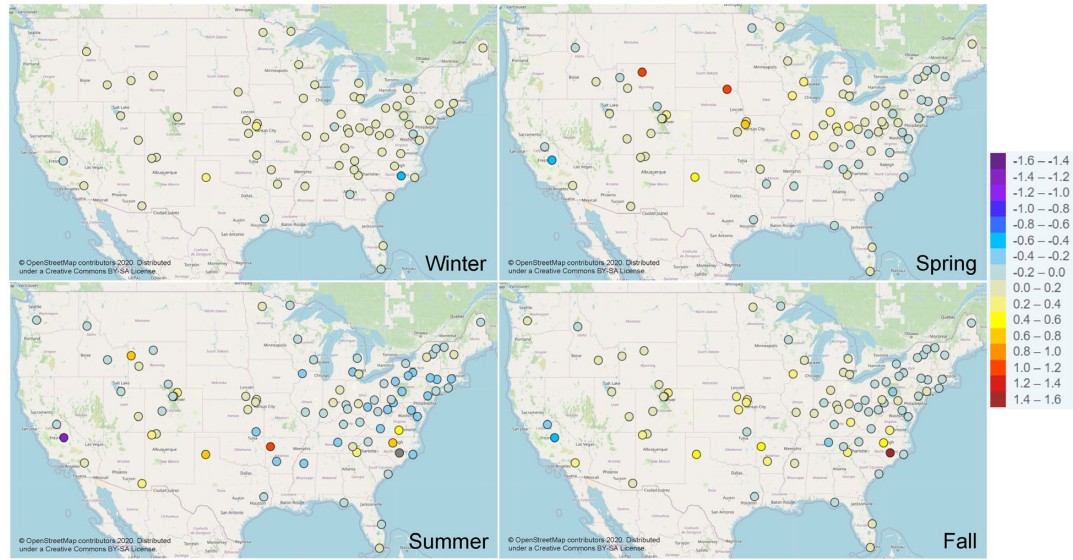

**Figure 13. Seasonal average NH₃ absolute bias difference (µg m⁻³) between CMAQ531_WRF411_M3Dry_BiDi and CMAQ531_WRF411_STAGE_BiDi simulations for all AMON sites. Cool shading (negative values) indicate smaller bias in the CMAQ531_WRF411_M3Dry_BiDi simulation, warm colors (positive values) indicate smaller bias in the CMAQ531_WRF411_STAGE_BiDi simulation. Grey shading indicates values beyond the range of the scale.**



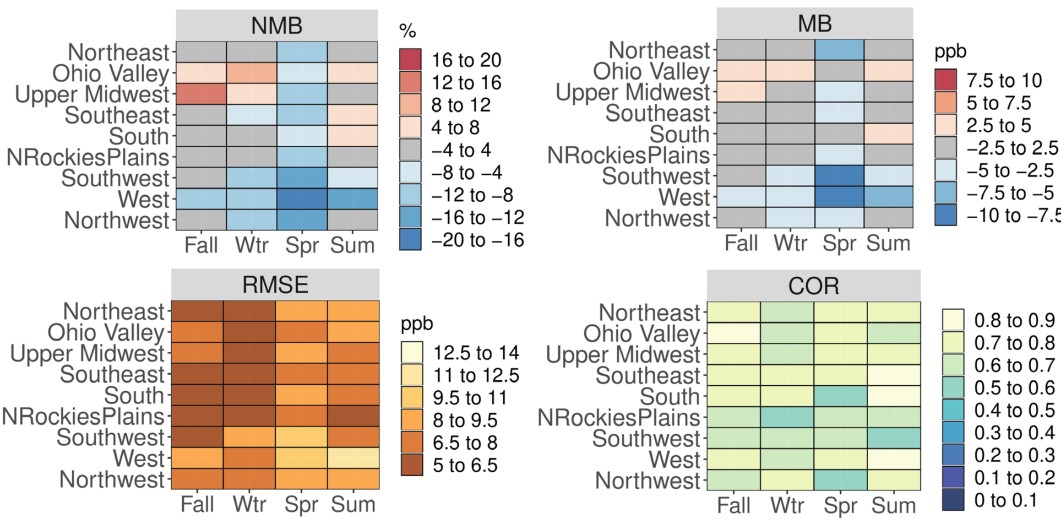

Figure 14. NMB (%), MB (ppbv), RMSE (ppbv), and Pearson correlation coefficient values for MDA8 O₃ for all AQS sites based on season and NOAA climate region for the CMAQ531_WRF411_M3Dry_BiDi simulation.

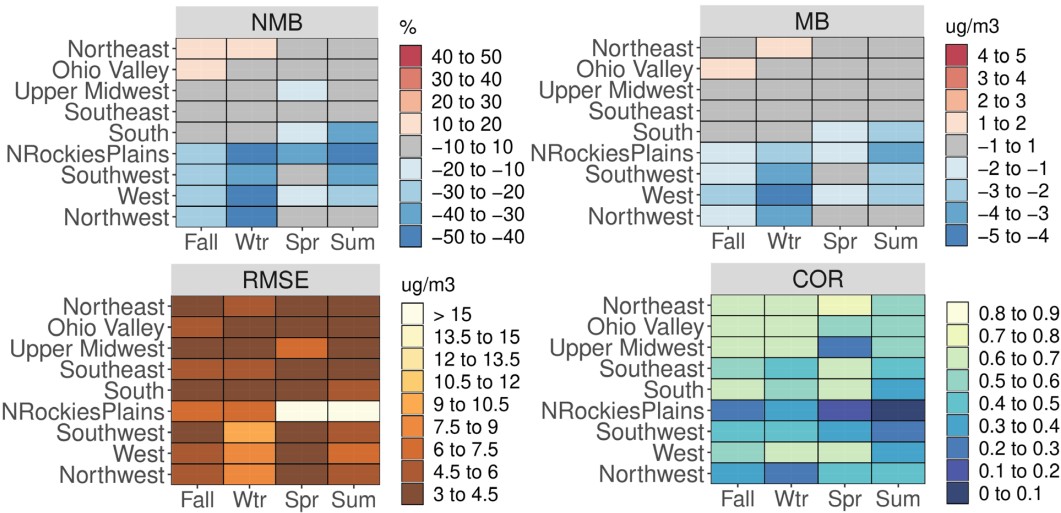

Figure 15. NMB (%), MB (µg m⁻³), RMSE (µg m⁻³), and Pearson correlation coefficient (COR) values for daily average PM₂.₅ for all AQS sites based on season and NOAA climate region for the CMAQ531_WRF411_M3Dry_BiDi simulation.