# Peer review of "The Community Multiscale Air Quality (CMAQ) Model Versions 5.3 and 5.3.1: System Updates and Evaluation"

_Geoscientific Model Development, 2020_

## Referee Comment (RC1) · Anonymous Referee #1 · 4 Nov 2020

This manuscript provides a description and evaluation of the newest versions of the CMAQ modeling system. The paper is extremely well-written, comprehensive and thorough. For the first time in my 30+ year career, I can find nothing to criticize or comment on. I recommend that the paper be published as is. Well done authors!

---

## Referee Comment (RC2) · Anonymous Referee #2 · 5 Jan 2021

Overview:

The paper (i) documents the model changes introduced in CMAQ versions 5.3 (and 5.3.1) compared to the previous version 5.2 and (ii) presents the evaluation of mean daily 8-hr average (MDA8) surface ozone, average PM2.5 and PM2.5 speciation over the USA. Hemispheric and regional model configuration of 5.2 and various variants of 5.3 are included in the evaluation.

General comments:

Peer-reviewed publication on model updates should be of interest to the scientific community, also beyond the specific user community of that model. These papers are Printer-friendly version

expected to differ from other types of documentation such as user-guides and comprehensive documentation because they should follow a certain scientific narrative.

The presented paper is generally a welcome and well-written overview of the CMAQ updates of version 5.3. However, the balance between a readable scientific paper and a user-guide presenting all possible configuration options was too often shifted to the latter, which made it sometimes difficult to understand the main points of the paper.

I strongly recommend making a more selective and clearer choice of the discussed 5.3 and 5.3.1 configuration options. A single 5.3. default configuration (it could also be 3.5.1) should be more clearly identified and compared against the default version 5.2. The subvariants of 5.3 and 5.3.1 can then be used to discuss further issues or to better substantiate the findings of the paper. Table 1 is central to the paper and should be introduced much earlier and it should be a more verbose.

Further, it should be better distinguished between 5.2 to 5.3 updates that change scientific result and technical updates of the modelling infrastructure such as MCIP or DESID, whose impact is not evaluated. I recommend shortening the description of these technical modifications, if their impact is not assessed in the paper.

The presented model run versions do often vary in more than one aspect, which makes pin-pointing the reasons for the differences in the model performance difficult. It is common practise that developments of comprehensive modelling systems are often implemented in "packages" containing various modification but a more detailed discussion on the dominating reasons for differences in the modelled concentrations and fluxes would be helpful. For example, a novelty aspect of STAGE compared to M3Dry seems to be the multi-tile approach. However, the dry deposition simulations seem to differ also in other aspects and a clear message on the importance of the tiled approach is not provided.

A discussion of the representativeness of the observations is required for scientifically sound evaluation paper. For AQ application it is common practise to distinguish differ-

GMDD
ent air quality regimes such as "urban", "rural" and "street". While representativeness is mentioned it is not acted upon as no stratification of the observations is applied in the many time series plots.

The evaluation of the model runs is limited to MDA8 ozone and seasonal averaged and speciated PM2.5. While these quantities might be the most important for air quality legislation purposes, the scientific validity of the paper could be greatly improved by also presenting the impact of the model changes on the diurnal cycle and simulated maximum values. Also including NOx and SO2 evaluation results in the main paper would be very welcome to understand the discussed changes in MDA8 ozone and PM2.5. If the model upgrade to 5.3 did not noticeably modify the aforementioned variables, it should be pointed out more clearly.

Overall, version 5.3. seems to be only a small improvement in ozone over 5.2 with the strong caveat of the increased ozone spring time bias and the general degradation in California throughout the year. It would be interesting to discuss the reasons in more detail and to explain why 5.3. is considered to be an improvement.

It is an interesting result that the pollutants boundary conditions simulated by the hemispheric configuration are one of the most influential aspects for the regional model differences. It would therefore also be useful to evaluate the hemispheric configuration with the AQS (or other) observations and not only show model differences between these simulations. Likewise, the performance of the regional and hemispheric runs should be assessed and compared.

Specific comments:

L 25ff The text seems to jump between version 5.3. and 5.3.1 and it is not clear to which you refer (see my general comment on a clearer 5.3 version choice)

L 27 Please quantify better the bias changes (x % or y ppb)

L 32 as above for PM.
L 43 Please mention if the tiling approach contributed to these differences.

L 93-99 I recommend referring already here to table 1 or insert a new overview table of the various model modification, which you will discuss in that section. Also make much clearer, which of the following updates are used only in the hemispheric and the regional configuration. It took a while to realise that the halogenic chemistry updates are not part of the regional model configuration, which is the focus of this paper.

L 115-129 It is not quite clear, if this refers to the 5.2 to 5.3 update or to a previous update.

L 124 How far inland there is a noticeable difference?

L 144 Is this loss rate only applied at the lowest model level or throughout the atmosphere?

L 163 The difference between AERO7 and AERO7i is not clear and if this is relevant to the paper. According to my understanding of Table 1 only AERO7 is used (?). Why do you mention 7i here?

L 176 "half organic" or "half of secondary organic" aerosol ?

L 200 Please explain "biogenic mapping"

L 194 -203 I find this a bit confusing. Please only discuss configuration options you are going to assess in the paper.

L 211 Please provide reference or explanation, why the coarse mode dry deposition velocities were considered to be too high in 5.2

L 216 Please explain "in areas of high sigma". Is the modification in 5.3 a numerical bugfix or does it imply a modification of the algorithm?

L 225 ff Again, it is not always clear which aspects of the aqueous and heterogenous chemistry are modified between 5.2. and 5.3 and if you test any in the paper.
L249-257 The reported changes in SO4 and O3 are substantial. It is not clear if they are derived from the represented simulation. If anything, they should be included in the results section 4.

L 270 FF: The main update of M3Dry seems to be the much more detailed input of NH3 information from fertilisation. It is not clear, if this is a 5.2 vs. 5.3 difference.

L 300 FF: Does STAGE use the M3Dry approach including its input data and differs it only w.r.t the sub-grid (tile) approach.

L 321 Please provide a reference or explanation for "bulk accommodation coefficient"

L 351-378 Please consider removing descriptive parts, which are more a user guide than information important for the conclusions of the peer-reviewed paper.

L 422 Here or elsewhere it be good to give more information about the kind of the impact of the different options for LAI and VF input into WRF on meteorological model results relevant for CMAQ such 2mT, PBL height or soil moisture.

L 441 Please provide more detail on the EPA wildland fire emissions for the regional runs. Has FINN been used?

L480 Table 1 needs to be introduced much earlier in the paper because it is the "lifeline" for the reader to understand the different tested model variants. Also, I strongly recommend to be more descriptive in the table (e.g if halogenic chemistry was used) and not only provide abbreviations.

L495 Using WBD, or not, needs further discussion. Why was WBD considered acceptable for the hemispheric runs and not for the regional runs ? What were the negative implications of not using WBD in areas affected by dust ?

L 510 Please clarify if you stratified the data in AQ regimes such as urban, rural and street ? An evaluation of an AQ model at 12 km resolution may not be representative for locally highly polluted areas. A discussion of the representativeness of the obser-

GMDD
vations is required for scientifically sound paper.

L 543 Please clarify, if the statistical measures are calculated for each station by meaning over time or also for specific times meaning over space.

L 578 I think showing the mean values (as in Fig S2 and S6) and not the biases (Fig1 and 3) would be a better choice for the paper. Please comment on the reason for the spring time underestimation in both 5.3 and 5.2

L587 The differences in ozone are attributed to changes in dry deposition. It seems that differences in meteorology (WRF38 vs WRF 411) also impact the mean ozone values. WRF411 ozone is about 1ppb higher than WRF38. Please provide more detail what the reasons are for this increase in ozone.

L 593 As the degradation in spring is striking, please provide more information why this is the case.

L 600 As the degradation in California is striking, please provide more information why this is the case.

L 611 The large summer underestimation is a common feature in 5.3 and 5.2 (see Fig S6). Please comment, why this could be the case.

L 615 Please provide more details on the impact of OH and Ozone in winter on OC. What are the main pathways? Why does the opposite ozone difference (i.e. higher O3 on 5.2) during the rest of the year do not lead to higher PM2.5 in 5.2 ?

L 616 Please specify in more detail what modelled species NCOM entails.

L 636 I find this explanation (different RWC emission in Canada) confusing because I had assumed the emissions are the same for both 5.2 and the considered 5.3 run. Please comment and point out this fundamental difference earlier. I assume that PC-SOA on RWC is not applied in the presented case.

L 657 Please provide more information what species are included in the OTHR group

GMDD
of the model results.

L 670 Please comment why BELD5 has been used despite that it is not publicly available and seem to degrade the PM results in summer, when the model is already biased low.

L 675-600 Please also provide (a) an evaluation of the HCMA runs with observations and (b) compare the performance of the regional and hemispheric runs (see general comment). Please mention the ozone sonde evaluation in the supplement, which shows larger negative biases with 5.3 in the troposphere.

L 712: Please clarify, if the difference between WRF411 and WRF38 runs also includes the differences in vegetation parameters (MODIS based) for dry deposition and biogenic emission modelling, or not.

L730-742 Please make clearer if this paragraph is meant to be the explanation for the differences mentioned in the paragraph before. It would also be helpful to stress more (if it is the case) that the differences in ozone are mainly driven by the vegetation data updates not by other WRF changes.

L 740 Is the reduced O3 wet scavenging something quantified by model diagnostics or is it more a guess. One would not expect a large impact of wet deposition, especially when aerosols seemed to be not effected (see next paragraph).

L 759 Please summarise here the main differences. An important difference of STAGE seems the tile-approach but it seems to make no differences at all as it is not mentioned in the section. Please remark more clearly if that aspect has an (positive) impact.

L 778 Please discuss the reasons for the differences in ozone based on the differences between M3Dry and STAGE.

L 796 The NH3 evaluation is a very interesting aspect of the paper. I strongly recommend to also show in Fig 12 a run of the noBIDI configuration to demonstrate the impact of the bi-directional approach. Fig 12 is very busy and I suggest considering
separate plots for the different error measures.

L 805: It would be interesting to know why the differences are so large at a few particular stations shown in Fig 13. whereas the differences are much smaller for the majority of the stations. Is this the tile -approach or other issues related to vegetation cover?

L 813-850 Figures 14 and 15 provide a very welcome overview of the model performance. I would strongly recommend to prepare the graphs not only for 5.3 but also for 5.2. In that case the reader would get the possibility to compare the performance of the two configurations, which is after all the main topic of the paper.

L 881 Please mention the specific differences between STAGE and M3Dry that cause the described differences in the results.

Table 1: Please make it more self-contained, i.e. readable without need to refer to the paper for each acronym. Please add information if the run is evaluated with observations. Please consider to chose a 5.3 runs as the default 5.3 version.

Figure 1/3: Please mention which lines are on top of each other. It is not clear where the blue line is. In my opinion, there is no need to repeat the color-code information of the legend. Refer to table 1 for the different options.

Figure 2/4: Please mention briefly why the specific 5.3 version is chosen (5.3 default ?) Refer to table 1 for details.

Fig 5: Spell out unfamiliar acronyms such as OTHR and NCOM.

Fig 6/7: Say this are surface values

Fig 8-11: consider saying more clearly in the caption what aspect is different (i.e. meteorology or deposition model)

Fig 12: Consider adding a model run without bi-directional-flux approach. Do not show to many statistical parameters in one plot. Please clarify if the correlation is spatial or temporal.

GMDD
Fig 14/15: Please include and juxtapose the graphs for a 5.2 and 5.3 CMAQ simulation.

---

## Author Comment (AC1) · 11 Feb 2021

Response: We thank the Reviewer for the encouragement and high praise on our manuscript. We are flattered.

---

## Author Comment (AC2) · 11 Feb 2021

Interactive comment on "The Community Multiscale Air Quality (CMAQ) Model Versions 5.3 and 5.3.1: System Updates and Evaluation" by K.AppelWyat Appel et al. Anonymous Referee #2 Overview: The paper (i) documents the model changes introduced in CMAQ versions 5.3 (and 5.3.1) compared to the previous version 5.2 and (ii) presents the evaluation of mean daily 8-hr average (MDA8) surface ozone, average PM2.5 and PM2.5 speciation over the USA. Hemispheric and regional model configuration of 5.2 and various variants of 5.3 are included in the evaluation.

General comments: Peer-reviewed publication on model updates should be of interest to the scientific community, also beyond the specific user community of that model. These papers are expected to differ from other types of documentation such as user-guides and comprehensive documentation because they should follow a certain scientific narrative. The presented paper is generally a welcome and well-written overview of the CMAQ updates of version 5.3. However, the balance between a readable scientific paper and a user-guide presenting all possible configuration options was too often shifted to the latter, which made it sometimes difficult to understand the main points of the paper. I strongly recommend making a more selective and clearer choice of the discussed 5.3 and 5.3.1 configuration options. A single 5.3. default configuration (it could also be 3.5.1) should be more clearly identified and compared against the default version 5.2. The subvariants of 5.3 and 5.3.1 can then be used to discuss further issues or to better substantiate the findings of the paper. Table 1 is central to the paper and should be introduced much earlier and it should be a more verbose. Further, it should be better distinguished between 5.2 to 5.3 updates that change scientific result and technical updates of the modelling infrastructure such as MCIP or DESID, whose impact is not evaluated. I recommend shortening the description of these technical modifications, if their impact is not assessed in the paper. The presented model run versions do often vary in more than one aspect, which makes pin-pointing the reasons for the differences in the model performance difficult. It is common practise that developments of comprehensive modelling systems are often implemented in "packages" containing various modification but a more detailed discussion on the dominating reasons for differences in the modelled concentrations and fluxes would be helpful. For example, a novelty aspect of STAGE compared to M3Dry seems to be the multi-tile approach. However, the dry deposition simulations seem to differ also in other aspects and a clear message on the importance of the tiled approach is not provided.

Response: We appreciate the Reviewer's perspective and suggestions for organizing the paper and for selecting the content to emphasize. Indeed, it is a tremendous challenge to balance the development details, evaluation, and analysis in such a
comprehensive modeling update, while also mutually satisfying the need to prepare a bonafide research article. Given that we cannot publish articles of unlimited length, the author team judiciously selected the depth of the descriptions and evaluation for this manuscript, which obviously limited the breadth of what we presented here. Some of the details that the Reviewer seeks will be self-contained in other forthcoming publications that isolate the focus on specific scientific updates to CMAQ.

Several changes were introduced in the revised manuscript to respond to some of the Reviewer's general comments. Section 4, which presents the evaluation results, has been reordered to present the comparison between CMAQv521 and a base CMAQv531 simulation, namely the CMAQ531_M3Dry_WRF411_BiDi simulation. The comparison between the two simulations (previously presented in section 4.5) is now presented in section 4.1. In addition, two new figures have been added to the main text to address the Reviewer's specific suggestions; these new figures present the CMAQv521 seasonal/regional statistics for MDA8 O3 and PM2.5 to complement the two figures that present the same results for the CMAQv531 simulation. The discussion has also been expanded to include a more detailed analysis of major PM2.5 component species (SO4, NO3, OC and EC) and NOX/SO2.

A discussion of the representativeness of the observations is required for scientifically sound evaluation paper. For AQ application it is common practise to distinguish different air quality regimes such as "urban", "rural" and "street". While representativeness is mentioned it is not acted upon as no stratification of the observations is applied in the many time series plots.

Response: As was mentioned above, we were limited in the breadth of analysis that could be presented succinctly in the manuscript. We examined the AQS data stratified by rural, suburban, and urban classification (the available classifications in the AQS data). The time series analysis of MDA8 O3 and PM2.5 by these classification regimes indicate similar observed and simulated values between all three regimes. Therefore, presenting all AQS data together represents the expected model performance (and
change in model performance). To address the Reviewer's comment, these additional comparisons by rural, suburban, and urban stratification have now been added to the supplement (Fig. S3), and a sentence was added to Sect. 4.2 indicating this result and pointing interested readers to the supplement.

The evaluation of the model runs is limited to MDA8 ozone and seasonal averaged and speciated PM2.5. While these quantities might be the most important for air quality legislation purposes, the scientific validity of the paper could be greatly improved by also presenting the impact of the model changes on the diurnal cycle and simulated maximum values. Also including NOx and SO2 evaluation results in the main paper would be very welcome to understand the discussed changes in MDA8 ozone and PM2.5. If the model upgrade to 5.3 did not noticeably modify the aforementioned variables, it should be pointed out more clearly.

Response: In response to the Reviewer's comment, seasonal diurnal plots of hourly ozone and PM2.5 have been added to the supplement (Figs. S9 – S16), and discussion of these plots has been added to Sect. 4.1. Following the reviewer's suggestion, a brief evaluation of NOx and SO2 (which did not change much between CMAQ521 and CMAQ531) has also been added to Sect. 4.1, with additional figures added to the supplement (Figs. S45 – S48).

Overall, version 5.3. seems to be only a small improvement in ozone over 5.2 with the strong caveat of the increased ozone spring time bias and the general degradation in California throughout the year. It would be interesting to discuss the reasons in more detail and to explain why 5.3. is considered to be an improvement.

Response: Sometimes improvements in the underlying model science do not drastically improve model performance in a bulk evaluation. However, those changes improve the representation of the atmosphere and its interactions with the ground, water, and various pollutants on temporal and spatial scales that are not illustrated as effectively in this paper. Because the majority of the updates in CMAQ531 focused

on aerosol chemistry, large differences in ozone were not expected. However, there was considerable improvement in wintertime ozone from combining multiple scientific changes. The springtime degradation in performance is a result of issues with under-estimated ozone in the boundary conditions, which has pointed to the need to further investigate and update the marine chemistry in CMAQ to improve ozone in the HC-MAQ simulations. Some of the targeted improvements with the scientific changes will be illustrated in forthcoming papers on those science components. It is worth noting that wintertime O3 bias improved substantially with CMAQv53.

It is an interesting result that the pollutants boundary conditions simulated by the hemispheric configuration are one of the most influential aspects for the regional model differences. It would therefore also be useful to evaluate the hemispheric configuration with the AQS (or other) observations and not only show model differences between these simulations. Likewise, the performance of the regional and hemispheric runs should be assessed and compared.

Response: Lateral BCs have always been known to have the potential to play a large role in the regional AQ simulations. Their relative importance has increased over time as the proportion of the contribution from transported and natural pollutants increases as the concentration of locally (regionally) produced pollutants has decreased. The authors agree that the accurate representation of BCs is becoming more important and will continue to do so as regional-scale pollution falls. Focused and continued evaluation of the hemispheric simulations and comparison to the regional simulation are underway and could be an interesting topic for a future manuscript.

Specific comments: L 25ff The text seems to jump between version 5.3. and 5.3.1 and it is not clear to which you refer (see my general comment on a clearer 5.3 version choice)

Response: The text has been updated to clarify that CMAQ531 is being referenced.

L 27 Please quantify better the bias changes (x % or y ppb)

Response: The bias change has now been quantified in the text.

L 32 as above for PM.

Response: The bias change has now been quantified in the text.

L 43 Please mention if the tiling approach contributed to these differences.

Response: Because of the scientific (formulation, parameters) differences between the M3Dry and STAGE models, it is not feasible to associate specific differences between them with differences in O3 and PM2.5; nor can the noted differences solely be attributed to the differences in tiling approach. Because the Reviewer has raised this point several times, a sentence was added to Sect. 4.5 to caution the reader against this interest in attributing difference in species behavior with specific aspects of either model.

L 93-99 I recommend referring already here to table 1 or insert a new overview table of the various model modification, which you will discuss in that section. Also make much clearer, which of the following updates are used only in the hemispheric and the regional configuration. It took a while to realise that the halogenic chemistry updates are not part of the regional model configuration, which is the focus of this paper.

Response: In response to the Reviewer, Table 1 is now referenced earlier in the manuscript.

L 115-129 It is not quite clear, if this refers to the 5.2 to 5.3 update or to a previous update.

Response: The text has been modified to indicate which updates were in CMAQ52 and which were in CMAQ53.

L 124 How far inland there is a noticeable difference?

Response: The inland impact is highly variable based on atmospheric conditions. As expected, the impact is typically larger closer to the coast.

L 144 Is this loss rate only applied at the lowest model level or throughout the atmosphere?

Response: Halogen mediated O3 loss is applied throughout the vertical layers. The rate constant is a function of atmospheric pressure and the values at lower layers (higher atmospheric pressures) are greater than the values at upper layers (lower atmospheric pressures). The text has been updated to reflect this detail.

L 163 The difference between AERO7 and AERO7i is not clear and if this is relevant to the paper. According to my understanding of Table 1 only AERO7 is used (?). Why do you mention 7i here?

Response: In the revised manuscript, this section has been shortened to highlight the key differences between AERO7 and AERO7i. Although all of the simulations evaluated in the manuscript use AERO7, the introduction to AERO7i is included for applications that require detailed isoprene chemistry.

L 176 "half organic" or "half of secondary organic" aerosol?

Response: Half the total organic aerosol, as originally stated in the text.

L 200 Please explain "biogenic mapping"

Response: As requested by the Reviewer, the text has been revised to rewrite biogenic mapping as "species to mechanism surrogate mapping".

L 194 -203 I find this a bit confusing. Please only discuss configuration options you are going to assess in the paper.

Response: As can be gleaned from the title of the paper and the Introduction, this article aims to both evaluate the CMAQ531 modeling system and discuss the major scientific updates to the modeling system. We feel it is worthwhile to describe the major scientific updates, even if they are not evaluated specifically, since those updates are targeted for other applications with CMAQ. Throughout the manuscript, we further

clarify which options are evaluated in the manuscript and which are not.

L 211 Please provide reference or explanation, why the coarse mode dry deposition velocities were considered to be too high in 5.2

Response: A separate manuscript (led by one of the authors) is under development that details these changes.

L 216 Please explain "in areas of high sigma". Is the modification in 5.3 a numerical bugfix or does it imply a modification of the algorithm?

Response: This is an algorithm update, as was originally stated in the text, to adjust for unrealistically high coarse mode particle deposition. To improve clarity and minimize confusion, the clause "in areas of high sigma" was removed from the sentence.

L 225 ff Again, it is not always clear which aspects of the aqueous and heterogenous chemistry are modified between 5.2. and 5.3 and if you test any in the paper.

Response: Sect. 2.1.7 states that two additional KMT cloud chemistry options were added to CMAQ53, namely KMT2 and KMTBR. These are the new KMT cloud chemistry options available in CMAQ53. The last sentence of Sect. 2.1.7 indicates how KMT affected the simulations that were evaluated in the manuscript. That sentence was revised to improve clarity.

L249-257 The reported changes in SO4 and O3 are substantial. It is not clear if they are derived from the represented simulation. If anything, they should be included in the results section 4.

Response: The reported SO4 and O3 changes are for the KMT2 implementation which was not used in the simulations presented Sect. 4. As mentioned at the end of Sect. 2.1.7 (where this analysis is shown), KMT2 is a research-grade option, and it was not used in the simulations that were evaluated in Sect. 4.

L 270 FF: The main update of M3Dry seems to be the much more detailed input of

NH3 information from fertilisation. It is not clear, if this is a 5.2 vs. 5.3 difference.

Response: The text has been modified to indicate that this update is specific to CMAQ53.

L 300 FF: Does STAGE use the M3Dry approach including its input data and differs it only w.r.t the sub-grid (tile) approach.

Response: No. As stated in our response to earlier comments and also summarized briefly in the manuscript, there are several differences in the science between STAGE and M3Dry, and those differences are not only related to tiling.

L 321 Please provide a reference or explanation for "bulk accommodation coefficient"

Response: The text has been modified to the more common terminology "mass accommodation coefficient" and a reference to Fahey et al., (2017) has been added.

L 351-378 Please consider removing descriptive parts, which are more a user guide than information important for the conclusions of the peer-reviewed paper.

Response: The section describing the DESID module has been significantly shortened, and reference to Murphy et al. (2020) has been added.

L 422 Here or elsewhere it be good to give more information about the kind of the impact of the different options for LAI and VF input into WRF on meteorological model results relevant for CMAQ such 2mT, PBL height or soil moisture.

Response: While that information would be useful in this context, it remains unpublished. We believe that adding that analysis of WRF to this manuscript could be considered tangential to our focus on updates and evaluation of CMAQ5.3. This information is available in previously published articles, references for which are now provided in the updated text.

L 441 Please provide more detail on the EPA wildland fire emissions for the regional runs. Has FINN been used?

Response: SMARTFIRE was used for U.S. fires and FINN was used for non-U.S. fires. The text has been updated to reflect this detail.

L480 Table 1 needs to be introduced much earlier in the paper because it is the "lifeline" for the reader to understand the different tested model variants. Also, I strongly recommend to be more descriptive in the table (e.g if halogenic chemistry was used) and not only provide abbreviations.

Response: Table 1 has been updated to include the halogen/DMS chemistry information. However, the authors have chosen to retain the acronyms for brevity.

L495 Using WBD, or not, needs further discussion. Why was WBD considered acceptable for the hemispheric runs and not for the regional runs? What were the negative implications of not using WBD in areas affected by dust?

Response: To account for transcontinental transport of PM2.5 from WBD (e.g., Sahara Desert), the WBD option was implemented in the hemispheric simulations, regardless of the potential for overestimations of PM2.5 from WBD. Given the rather small overall contribution to PM2.5 from WBD over the CONUS, the authors opted to forgo implemented WBD. A statement has been added to the explaining why WBD was used in the hemispheric simulations and not the CONUS.

L 510 Please clarify if you stratified the data in AQ regimes such as urban, rural and street ? An evaluation of an AQ model at 12 km resolution may not be representative for locally highly polluted areas. A discussion of the representativeness of the observations is required for scientifically sound paper.

Response: This was addressed under the Reviewer's General Comment. We disagree with the Reviewer's assertion that a discussion of the representativeness of observations is required for a scientifically sound paper. Several scientifically sound studies have been conducted and published without that line of analysis.

L 543 Please clarify, if the statistical measures are calculated for each station by mean-

ing over time or also for specific times meaning over space.

Response: Each statistical measure is calculated based on the individual model/observation pair (paired in both space and time) and then averaged spatially and/or temporally for each plot. Spatial plots are averaged temporally (to create seasonal averages) while time series plots are averaged both spatially and temporally to create monthly average values for all sites.

L 578 I think showing the mean values (as in Fig S2 and S6) and not the biases (Fig1 and 3) would be a better choice for the paper. Please comment on the reason for the spring time underestimation in both 5.3 and 5.2

Response: The plots show the difference in bias between the two simulations, not simply the bias. Therefore, the plots show which model has smaller/larger bias. The mean value plots shown in the supplement can illustrate whether the model is underpredicting or overpredicting a species. The plots in the main text demonstrate how the bias changes with the model updates. Since the primary goal of the article is to highlight the change in overall model performance between CMAQ521 and CMAQ531, the bias difference plots are showcased since they convey two pieces of information in a single plot.

L587 The differences in ozone are attributed to changes in dry deposition. It seems that differences in meteorology (WRF38 vs WRF 411) also impact the mean ozone values. WRF411 ozone is about 1ppb higher than WRF38. Please provide more detail what the reasons are for this increase in ozone.

Response: There were many updates between WRF38 and WRF411 that could impact the meteorology that was used in this evaluation. Unfortunately, it's well beyond the scope of this paper to associate specific updates in WRF411 with changes to the AQ species; that could be an entire study on its own. The hybrid vertical coordinate in WRF411 could have a relatively large impact in areas of complex terrain, and updates to the vegetation parameters in WRF411 also will induce localized changes. These

were identified in Sect. 3.1 as likely to impact the CMAQ results.

L 593 As the degradation in spring is striking, please provide more information why this is the case.

Response: It now stated up front that the impact in spring is primarily driven by lower O3 mixing ratios advected to the domain from the lateral BCs. With the text rearranged, this should be clearer now.

L 600 As the degradation in California is striking, please provide more information why this is the case.

Response: The text states that the O3 is generally underestimated in southern California in the Spring and Summer, and that the broad reduction in ambient O3 mixing ratios increases this underestimation. Text has been added to state that this reduction in ambient O3 mixing ratios is primarily due to the increased O3 dry deposition in CMAQ53.

L 611 The large summer underestimation is a common feature in 5.3 and 5.2 (see Fig S6). Please comment, why this could be the case.

Response: It is noted in the text that the summer underestimation is driven primarily by an underestimation of the "other" mass (see stacked bar plots in Fig. 9). If one were to remove the "other" mass from the analysis, CMAQ would actually overestimate the sum of the other PM2.5 constituents in the summer. So, there is a need to better identify the measured "other" mass to identify the species in the model that are underestimated. This is now noted in the text as well.

Also worth noting is that assumptions (i.e. average OM:OC ratio) are made to estimate the observed "NCOM", and that uncertainties exist in that estimate which could also impact the observed OTHR. Put differently, in the observed stacked bar, both OTHR and NCOM are estimated, not directly measured, so conceivably "true" NCOM may be lower than what is shown in the plot, changing our conclusion about OTHR being

the driver of the model underestimation. NO3 and NH4 also seem to contribute to the summer underestimation, though they can't fully account for the gap in the total mass.

L 615 Please provide more details on the impact of OH and Ozone in winter on OC. What are the main pathways? Why does the opposite ozone difference (i.e. higher O3 on 5.2) during the rest of the year do not lead to higher PM2.5 in 5.2 ?

Response: Higher O3 should generally lead to and be associated with higher OC as ozone itself oxidizes SOA precursors and is an indicator of OH oxidant abundance. SOA is produced throughout the year. In the winter, monoterpene oxidation (via O3, OH, and NO3) will produce SOA and the yields through all monoterpene pathways increased in v5.3 compared to v5.2. In winter, anthropogenic POA (e.g., from vehicles) as well as anthropogenic VOCs (e.g., pcSOA pathway and single-ring aromatics) will also oxidize via OH to SOA. The large wintertime overestimation noted in Sect. 4.2 likely came from erroneously using residential wood combustion (RWC) emissions with the pcSOA formulation. As a result, a large, artificial source of SOA was inserted, primarily in winter. The ability to remove that artificial source of SOA was one of the CMAQ53 updates. Simulations that do not include this artificial source of SOA are tagged "no RWC" in Table 1 to indicate that pcSOA pathway was not applied to the RWC emissions. The text has been modified to indicate that SOA from monoterpene oxidation is the primary pathway for increased OC in winter.

L 616 Please specify in more detail what modelled species NCOM entails.

Response: Non-carbon organic matter (NCOM) is Organic Matter minus Organic Carbon. Readers can refer to CMAQ documentation to get further details regarding the specific model species that make up NCOM. The parenthetical description of NCOM has been updated to improve clarity.

L 636 I find this explanation (different RWC emission in Canada) confusing because I had assumed the emissions are the same for both 5.2 and the considered 5.3 run. Please comment and point out this fundamental difference earlier. I assume that PC-

[Figure]

SOA on RWC is not applied in the presented case.

Response: The text has been modified to clarify that the Canadian emissions did not have RWC emissions separated from the other gridded emissions (unlike the U.S. inventory where RWC was separated from the other gridded emissions). And since PCSOA was applied in the simulations, the Canadian emissions would have had PC-SOA applied to RWC emissions whereas the U.S. emissions would not. This results in anomalously high PM2.5 for the CMAQ531 simulations. The modified text now makes this point clearer.

L 657 Please provide more information what species are included in the OTHR group of the model results.

Response: The other species in the model simply consists of the portion of primary emitted PM mass for which no detailed speciation information is available. Therefore, no specific species are attributed to other. For further details, see https://www.airqualitymodeling.org/index.php?title=CMAQ_version_5.0_%28February_2012_release%29_Technical_Docu

L 670 Please comment why BELD5 has been used despite that it is not publicly available and seem to degrade the PM results in summer, when the model is already biased low.

Response: The BELD5 data, which represents more accurate land-use information, are now available and the text has been updated to indicate this detail. The degradation in the summer is the likely the result of compensating bias in the modeling system.

L 675-600 Please also provide (a) an evaluation of the HCMA runs with observations and (b) compare the performance of the regional and hemispheric runs (see general comment). Please mention the ozone sonde evaluation in the supplement, which shows larger negative biases with 5.3 in the troposphere.

Response: Unfortunately, including a detailed evaluation of the HCMAQ simulations is simply beyond the scope of this manuscript and would greatly increase the length of

what is already a long manuscript. We provided information relevant to the impact that the differences in the HCMAQ ozone and PM2.5 concentrations have on the BCs used in the CONUS simulations. The comparisons against ozonesondes are now mentioned in section 4.1 to further indicate the lower O3 throughout the troposphere at many sites in the HCMAQ53 simulation.

L 712: Please clarify, if the difference between WRF411 and WRF38 runs also includes the differences in vegetation parameters (MODIS based) for dry deposition and biogenic emission modelling, or not.

Response: The relevant differences between the two WRF versions is described in Sect. 3.1. The MODIS-based vegetation parameters are used in WRF411, so they will directly impact the dry deposition and indirectly affects biogenic emission modeling by changing temperature, precipitation, soil moisture, etc. The text in question has been updated to indicate that WRF411 includes "updated land-surface parameters" in addition to HVC.

L730-742 Please make clearer if this paragraph is meant to be the explanation for the differences mentioned in the paragraph before. It would also be helpful to stress more (if it is the case) that the differences in ozone are mainly driven by the vegetation data updates not by other WRF changes.

Response: As was stated in the text, the differences are indeed primarily due to reduced dry deposition of O3 to vegetation. The text has been modified to provide a better segue between the paragraphs.

L 740 Is the reduced O3 wet scavenging something quantified by model diagnostics or is it more a guess. One would not expect a large impact of wet deposition, especially when aerosols seemed to be not affected (see next paragraph).

Response We agree with the reviewer that wet removal is not a dominant sink of tropospheric O3. The discussion was intended to point out that the lower precipitation in

[Figure]

WRF411 results in less O3 wet scavenging and may also influence the noted changes in O3 values.

L 759 Please summarise here the main differences. An important difference of STAGE seems the tile-approach but it seems to make no differences at all as it is not mentioned in the section. Please remark more clearly if that aspect has an (positive) impact.

Response: The main differences between M3Dry and STAGE were discussed in section 2.2. The tiling aspect of STAGE is among the differences between the two models, but attributing a specific sensitivity to the tiling is not possible given the breadth of differences between M3Dry and STAGE.

L 778 Please discuss the reasons for the differences in ozone based on the differences between M3Dry and STAGE.

Response: Unfortunately, the numerous scientific differences between the M3Dry and STAGE models limit our ability to attribute changes in ozone to a specific difference between M3Dry and STAGE.

L 796 The NH3 evaluation is a very interesting aspect of the paper. I strongly recommend to also show in Fig 12 a run of the noBIDI configuration to demonstrate the impact of the bi-directional approach. Fig 12 is very busy and I suggest considering separate plots for the different error measures.

Response: We appreciate the suggestion to augment Fig. 12. However, noBiDi simulations were not conducted for both the STAGE and M3Dry configurations. The goal of the STAGE/M3Dry comparison is to present the differences from applying each deposition model. Analyses of the impact of using BiDi have been previously presented in the individual M3Dry/STAGE manuscripts referenced in the paper.

L 805: It would be interesting to know why the differences are so large at a few particular stations shown in Fig 13. whereas the differences are much smaller for the majority of the stations. Is this the tile -approach or other issues related to vegetation cover?

Response: That is an interesting question, but we chose not to address that here because the manuscript is already quite long and to answer the question thoroughly would likely require significant additional analysis. Follow-on articles from the team will likely investigate the STAGE/M3Dry differences for NH3 in more detail, with particularly attention paid to those sites where differences are large. Such an analysis of differences at individual sites is better suited for that type of focused article.

L 813-850 Figures 14 and 15 provide a very welcome overview of the model performance. I would strongly recommend to prepare the graphs not only for 5.3 but also for 5.2. In that case the reader would get the possibility to compare the performance of the two configurations, which is after all the main topic of the paper.

Response: This is a good suggestion. Similar figures to Figs. 14 and 15 (now Figs. 2 and 4) have now been added for CMAQ521 in the main text (Figs. 1 and 3) to provide a comparison of against the CMAQ531 results.

L 881 Please mention the specific differences between STAGE and M3Dry that cause the described differences in the results.

Response: As we mentioned previously, the scientific differences between STAGE and M3Dry limit the ability to perform this analysis.

Table 1: Please make it more self-contained, i.e. readable without need to refer to the paper for each acronym. Please add information if the run is evaluated with observations. Please consider to chose a 5.3 runs as the default 5.3 version.

Response: The table has been updated to include information regarding the DMS/Halogen chemistry used. However, expanding all the acronyms would make for a quite unruly table. We do not specify a default version of CMAQ53 because the selection of scientific options is driven by different model applications.

Figure 1/3: Please mention which lines are on top of each other. It is not clear where the blue line is. In my opinion, there is no need to repeat the color-code information of

the legend. Refer to table 1 for the different options.

Response: The figure captions have been updated accordingly.

Figure 2/4: Please mention briefly why the specific 5.3 version is chosen (5.3 default ?) Refer to table 1 for details.

Response: The choice for the runs shown is provided in the text where the figures are first referenced. For figures 2/4, the simulations highlight the impact from science updates between CMAQ521 and CMAQ531. As such, the driving meteorology is the same for the two simulations (WRF38).

Fig 5: Spell out unfamiliar acronyms such as OTHR and NCOM.

Response: NCOM and OTHR are described in the text near the discussion of Fig. 9 (previously Fig. 5) is referenced.

Fig 6/7: Say this are surface values

Response: The figure captions have been updated to indicate the values are surface level.

Fig 8-11: consider saying more clearly in the caption what aspect is different (i.e. meteorology or deposition model)

Response: Text has been added to the figure caption to highlight the specific update being examined.

Fig 12: Consider adding a model run without bi-directional-flux approach. Do not show to many statistical parameters in one plot. Please clarify if the correlation is spatial or temporal.

Response: As discussed earlier, Fig. 12 was not augmented with another line for BiDi. We believe showing these statistical parameters concurrently paints a comprehensive and succinct image to compare these simulations. The correlation is both spatial and

temporal, as the correlation value is calculated for each model/ob pair and then averaged to produce a single correlation value.

Fig 14/15: Please include and juxtapose the graphs for a 5.2 and 5.3 CMAQ simulation.

Response: As requested by the Reviewer, analogous figures have been added for CMAQ521 as Figs. 1 and 3 of the revised manuscript.

---

## Author Response (AR2)

Topical Editor Decision: Publish subject to technical corrections (22 Mar 2021) by Samuel Remy
Comments to the Author:
Dear authors,

Please find below the last recommandations of reviewer #2:

"Please add legend to Figure S3, i.e. what lines (red or black) observations and model result.
Please add unit for deposition velocities to Figures S61 and 62
Please add meaning of circle size in Figures 54 and S55
I recommend a final check on all Figures to make sure units and legends are OK."

Could you please update your manuscript to take thos remarks into account? Many thanks in advance.

Kind regards,
Samuel

Author Response: The legends for figure S3 was amended to include line color specification. The legends in Figures S54 and S55 were updated to include the meaning of the symbol size, where symbol size is simply commensurate with the absolute value of the bias. The legends for Figures S61 and S62 were updated to include the units of deposition velocity. Several additional figures in the supplement were corrected for incorrect units in the legends. All figures in both the main text and supplement were checked for accuracy and no other errors were found.